# Not Just Object, But State: Compositional Incremental Learning without Forgetting

**Yanyi Zhang** [1], **Binglin Qiu** [1], **Qi Jia** [1], **Yu Liu** [*1], **Ran He** [2]

[1] International School of Information Science & Engineering, Dalian University of Technology
[2] MAIS&CRIPAC, Institute of Automation, Chinese Academy of Sciences
`yanyi.zhang@mail.dlut.edu.cn, m1andy@mail.dlut.edu.cn`
`jiaqi@dlut.edu.cn, liuyu8824@dlut.edu.cn, rhe@nlpr.ia.ac.cn`

## Abstract

Most incremental learners excessively prioritize coarse classes of objects while neglecting various kinds of states (*e.g.* color and material) attached to the objects. As a result, they are limited in the ability to reason fine-grained compositionality of state-object pairs. To remedy this limitation, we propose a novel task called **Compositional Incremental Learning** (composition-IL), enabling the model to recognize state-object compositions as a whole in an incremental learning fashion. Since the lack of suitable benchmarks, we re-organize two existing datasets and make them tailored for composition-IL. Then, we propose a prompt-based **Comp**osition **I**ncremental **Lear**n**er** (**CompILer**), to overcome the ambiguous composition boundary problem which challenges composition-IL largely. Specifically, we exploit multi-pool prompt learning, which is regularized by inter-pool prompt discrepancy and intra-pool prompt diversity. Besides, we devise object-injected state prompting by using object prompts to guide the selection of state prompts. Furthermore, we fuse the selected prompts by a generalized-mean strategy, to eliminate irrelevant information learned in the prompts. Extensive experiments on two datasets exhibit state-of-the-art performance achieved by CompILer. Code and datasets are available at: `https://github.com/Yanyi-Zhang/CompILer`.

## 1   Introduction

Class Incremental Learning (class-IL) [37, 22, 16, 10] gathers increasing attention due to its ability to make the models learn new tasks rapidly, without forgetting previously acquired knowledge. Yet, traditional class-IL sets a strict limit on the old classes such that they should not recur in newly incoming tasks. To break such a strict limitation, recent studies develop a new setting mostly called Blurry Incremental Learning (blur-IL) [24, 11], where the incremental sessions allow the recurrence of previous classes, resulting in a more realistic and flexible scenario. Despite such empirical progresses on incremental learning, they aim to improve object classification only, while overlooking fine-grained states attached to the objects. For instance, analyzing how the clothing styles (akin to states) have changed over time is important for forecasting the future trends that will emerge.

To simultaneously model objects and their states, some efforts are dedicated to Compositional Learning whose aim is how to equip the models with *compositionality* [4, 31, 45]. The core of compositional learning lies in the structure of class labels, which conceptualizes a state-object pair (*e.g.* "Brown Pants" and "Yellow Dress") as a whole, rather than a lonely object label. In this way, the model can dissect and reassemble learned knowledge, achieving a more fine-grained understanding about the objects. However, existing works are mainly focused on zero-shot generalization from seen

---

[*]corresponding author

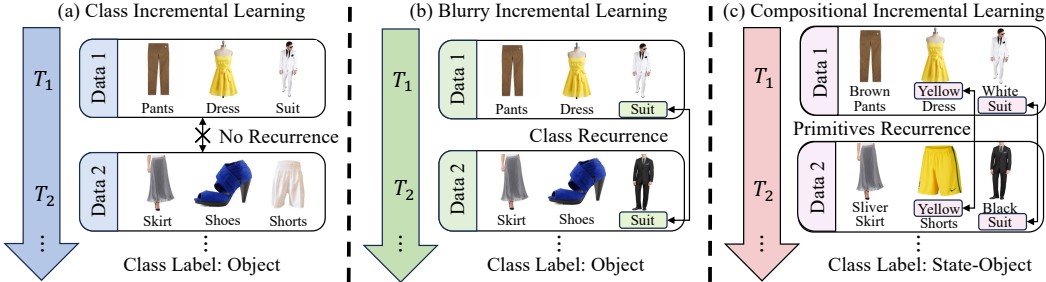

Figure 1: Differences between Class Incremental Learning (class-IL), Blurry Incremental Learning (blur-IL), and Compositional Incremental Learning (composition-IL). The object classes are not allowed to recur in the class-IL scenario, whereas they may recur randomly in the blur-IL scenario. Different from them, the classes in composition-IL involve state-object compositions apart from the object classes. Besides, the compositions do not reoccur, but the primitives (states or objects) may randomly reappear across incremental sessions.

compositions to unseen ones [26, 27, 5, 15], whereas none of them consider the challenging fact that the model must deal with a significantly larger number of composition classes than object classes. As a result, it is hardly feasible to learn all compositions by training the model once.

To remedy the limitations inherent in incremental learning and compositional learning, we conceive a novel task named **Compositional Incremental Learning** (composition-IL), enabling the model to continually learn new state-object compositions in an incremental fashion. As compared in Fig. 1, we can see that composition-IL integrates the characteristics of class-IL and blur-IL. Although the composition classes are disjoint across incremental tasks, the primitive classes (*i.e.* objects and states) encountered in old tasks are allowed to reappear in new tasks. Unfortunately, existing incremental learning approaches are challenged by such a compositional scenario, because their models excessively prioritize the object primitives while neglecting the state primitives. Consequently, the compositions with the same object but with different states become ambiguous and indistinguishable.

To tackle the problem, we propose a rehearsal-free and prompt-based **Comp**ositional **I**ncremental **Le**arner (**CompILer**). Specifically, our model comprises of three primary components: multi-pool prompt learning, object-injected state prompting, and generalized-mean prompt fusion. Firstly, we construct three prompt pools for learning the states, objects and compositions individually. Upon that, we add extra restrictions to regularize the inter-pool prompt discrepancy and intra-pool prompt diversity. This multi-pool prompt learning paradigm strengthens the fine-grained understanding and reasoning towards primitive concepts and their compositions. In addition, as the state classes are more difficult to distinguish than the object ones, we propose object-injected state prompting which incorporates object prompts to guide the selection of state prompts. Furthermore, we fuse the selected prompts by a generalized-mean fusion manner, which helps to adaptively eliminate irrelevant information learned in the prompts. Last but not least, we also leverage symmetric cross-entropy loss to alleviate the impact of noisy data during training.

In summary, the main contributions in this work are encapsulated as follows: (1) We devise a new task coined compositional incremental learning (composition-IL). It enables learning fine-grained state-object compositions continually while the isolated primitive concepts can randomly recur in incremental tasks. (2) To address the lack of datasets, we re-organize two existing datasets such that they are tailored specifically for composition-IL. For the two new datasets (Split-Clothing and Split-UT-Zappos), we split them into 5 and 10 incremental tasks for evaluating the methods. (3) We propose a novel learning-to-prompt model for composition-IL, namely CompILer. Our state-of-the-art results on Split-Clothing and Split-UT-Zappos validate the effectiveness of CompILer for incrementally learning new compositions without forgetting old ones.

## 2    Related Work

**Incremental Learning.** The approaches to addressing catastrophic forgetting for incremental learning can be broadly grouped into four categories: regularization based methods [6, 39] aim to protect

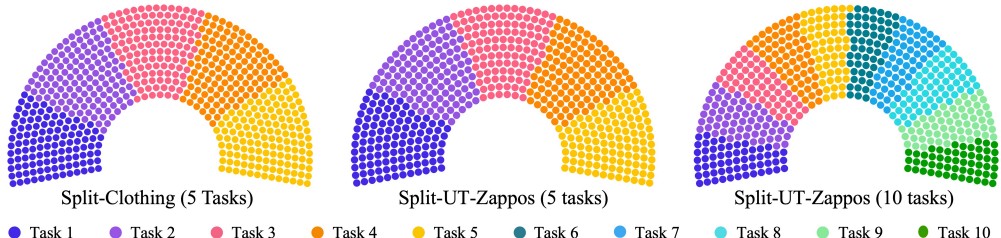

Figure 2: Data Statistics of Split-Clothing and Split-UT-Zappos for tasking composition-IL. Split-Clothing is divided into a 5-task scenario, while Split-UT-Zappos includes both 5-task and 10-task scenarios. In all settings, the number of images per task has been balanced properly.

influential weights of old experiences from updating; knowledge distillation based methods [16, 34] distill knowledge from the model trained on the previous tasks and adapt it to new tasks; rehearsal based methods [32, 48, 20] require a memory buffer to store some old data, so as to make the network remember previous tasks; parameter isolation methods allocates different model parameters to each task, to prevent any possible interference. Different from the methods, L2P [43] proposes an innovative learning-to-prompt paradigm, which incorporates plasticity and stability through adapting a set of learnable prompt tokens on top of a frozen pre-trained backbone. Inspired by L2P [43], more recent works [42, 36, 2, 28] take full advantage of various prompt tuning strategies, achieving new state-of-the-art performance for incremental learning. However, such methods take into account object classes solely, while neglecting various kinds of state classes associated with the objects. To this end, our work proposes compositional incremental learning with the purpose to continually identifying the composition classes of state-object pairs. Note that, Liao, *et al* [18] conduct an initial study toward the compositionality in incremental learning, whereas their attention is on the composition of multiple object classes (*e.g.* "Car" and "Person") in one image, rather than the state-object compositions in this work.

**Compositional Learning.** A major line of compositional learning research focuses on Compositional Zero-Shot Learning (CZSL) [23], which aims to infer unseen state-object compositions by acquiring knowledge from seen ones. Subsequent approaches building upon the CZSL setting further incorporate graph neural networks to model the dependency between primitives and compositions [25], and employ cosine classifiers to avoid being overly biased toward seen compositions [21]. Other approaches [13, 14, 12] propose training two classifiers to identify states and objects separately. The latest works [19, 3, 38, 8, 47] model both composition and primitives simultaneously, achieving state-of-the-art results. Albeit the numerous attempts made in compositional learning, they fail to consider an incremental learning paradigm given the increasing number of composition classes in open-world scenarios. Besides, directly applying CZSL methods to composition-IL might lead to a stale and decaying performance on forgetting. By contrast, our proposed CompILer markedly bypasses catastrophic forgetting with the help of multi-pool prompt learning.

## 3 Preliminaries

In this section, we firstly define the task of composition-IL, and then introduce two datasets we construct for the task, followed by revealing the ambiguous composition boundary problem.

### 3.1 Problem Definition

For composition-IL, a model sequentially learns $N$ tasks $\mathcal{T} = \{\mathcal{T}_1, \mathcal{T}_2, \cdots \mathcal{T}_N\}$ corresponding to a set of composition classes $\mathcal{C} = \{\mathcal{C}_1, \mathcal{C}_2, \cdots \mathcal{C}_N\}$. We note that the composition classes between incremental tasks are always disjoint, which means $\mathcal{C}_i \cap \mathcal{C}_j = \emptyset$ for any $i \neq j$. Different from the composition classes, the primitive classes are allowed to recur in different tasks. That means it allows the tasks to share some primitive concepts of objects and states. Therefore, we can define the set of all state and object classes with $\mathcal{S} = \{s_1, s_2, \cdots, s_n\}$ and $\mathcal{O} = \{o_1, o_2, \cdots, o_m\}$, respectively. Given each image $x$, it has a composition label $c$ which is constructed with a state label $s$ and an object label $o$, *i.e.* $c = <s, o>$, where $c \in \mathcal{C}$, $s \in \mathcal{S}$ and $o \in \mathcal{O}$. We take the example of "red shirt", where "red" is denoted with $s$, "shirt" corresponds to $o$, and "red shirt" is expressed with $c$.

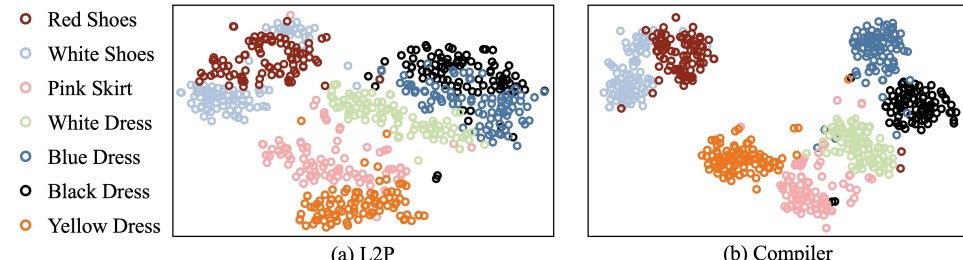

Figure 3: *t*-SNE feature distributions of seven compositions from the Split-Clothing benchmark. For the compositions with the same object but with different states, our CompILer achieves more distinguishable boundaries than the L2P baseline.

## 3.2 Dataset Construction

As there are no existing datasets suitable for composition-IL, we re-organize the data in Clothing16K [46] and UT-Zappos50K [44], and construct two new datasets tailored for composition-IL, namely **Split-Clothing** and **Split-UT-Zappos**. To be more specific, we firstly sort the composition classes based on the number of their images, and then select the foremost 35 compositions from Clothing16K and the top 80 from UT-Zappos50K, so as to construct Split-Clothing and Split-UT-Zappos, respectively. In this way, Split-Clothing encompasses 9 states and 8 objects while Split-UT-Zappos consists of 15 states and 12 objects in total. For Split-Clothing, we randomly partition the compositions into 5 tasks. Regarding Split-UT-Zappos, the compositions are sorted by count and are evenly divided into 5 and 10 tasks. The image distribution for each task is shown in Fig. 2. Note that, we elaborate more details on both datasets in the following technical appendix.

## 3.3 Revealing the Ambiguous Composition Boundary

The main stumbling block in composition-IL is the ambiguous composition boundary. Although the composition label consists of two primitives (*i.e.* object and state), we note that the model excessively prioritizes the object primitive while neglecting the state primitive. Consequently, the compositions with the same object but with different states become ambiguous and indistinguishable. To prove that, we apply L2P [43] to composition-IL, whereas it is challenged by significant ambiguities in composition classification. As illustrated in Fig. 3 (a), the *t*-SNE visualization showcases the entanglement among the compositions like "white dress", "black dress" and "blue dress". We conjecture that this ambiguous problem tends to become more severe when more tasks are arriving incrementally. To address it, we propose a new model namely CompILer, which disentangles compositions and primitives via a multi-pool prompt learning. Advantageously, our method promotes the learning on the states and establishes clearer composition boundaries, as shown in Fig. 3 (b).

## 4 Methodology

**Overview.** We leverage the learning-to-prompt paradigm [43] and develop a novel compositional incremental learner (CompILer) tailored specifically for composition-IL. As depicted in Fig. 4, CompILer comprises three primary components: multi-pool prompt learning, object-guided state prompting, and generalized-mean prompt fusion. Firstly, we initialize three prompt pools dedicated to learning and storing visual information related to states, objects and their compositions. In order to differentiate the knowledge learned across and within prompt pools, we define inter-pool discrepant loss and intra-pool diversified loss jointly. We then employ object prompts to guide the selection of state prompts, thereby improving the state representation learning. Moreover, we utilize a generalized-mean fusion to integrate the selected prompts in a learnable manner. Ultimately, we optimize the classification objective with symmetric cross-entropy loss, to alleviate the effect of noisy data.

### 4.1 Multi-pool Prompt Learning

The learning-to-prompt paradigm [43, 35], especially suitable for large pre-trained backbones, has opened up a new path for incremental learning. It has proven to incorporate plasticity and stability

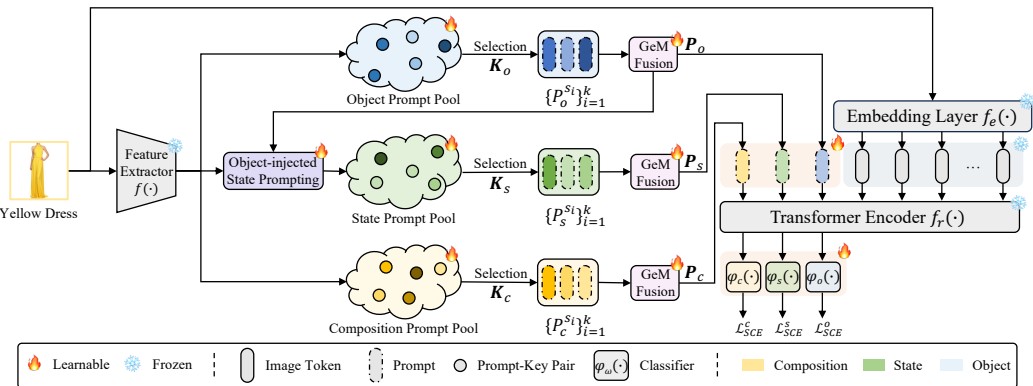

Figure 4: Overall architecture of our composition incremental learner (CompILer), which comprises multi-pool prompt learning, object-injected state prompting, and generalized-mean prompt fusion. The multi-pool prompt learning mechanism captures information related to states, objects, and their compositions, each through a dedicated pool. The object-injected state prompting utilizes the object prompt to promote the state representation learning. Moreover, the generalized-mean prompt fusion is used to prioritize the useful prompts and diminish the irrelevant ones.

better through adapting a set of learnable tokens in a prompt pool to a frozen pre-trained backbone. Nevertheless, existing prompt-based approaches are initially designed for class-IL, thereby building a single prompt pool for object classification solely. when dealing with state-object composition classification, they tend to excessively prioritize the object primitive while neglecting the state primitive. To this end, we propose to construct three discrepant and diversified prompt pools $\mathbb{P}_s$, $\mathbb{P}_o$ and $\mathbb{P}_c$, which serve to learn visual information related to states, objects and their compositions, respectively. Besides, each pool is associated with a set of learnable keys $\mathbb{K}_\omega$ for query-key prompt selection. The three prompt pools and their keys are defined as:

$$\mathbb{P}_\omega = \left\{ P_\omega^1, P_\omega^2, \cdots P_\omega^M \right\}, \mathbb{K}_\omega = \left\{ K_\omega^1, K_\omega^2, \cdots K_\omega^M \right\}, \omega \in \{s, o, c\}, \tag{1}$$

where $P_\omega^i \in \mathbb{R}^{L \times D}$ is a single prompt with token length $L$ and embedding dimension $D$. $K_\omega^i \in \mathbb{R}^D$, the key of $P_\omega^i$, is a learnable token with the same size. $M$ is the number of prompts in each pool.

One important concern in such multi-pool prompt learning is how to enrich the prompts with the avoidance of identical pools. To achieve it, we consider integrating **inter-pool prompt discrepancy** and **intra-pool prompt diversity** jointly. On the one hand, the inter-pool prompts should be discrepant as the visual information about states, objects, and compositions should be different. One the other hand, within each pool, the intra-pool prompts should be diversified so to capture more comprehensive features from all the classes.

In practice, we formulate a unified objective to regularize both inter-pool discrepancy and intra-pool diversity, by leveraging a simple and effective directional decoupled loss used in [17]. The directional decoupled (dd) loss between any two pools (e.g. $P_i$ and $P_j$) is formulated as:

$$\mathcal{L}_{dd}^{(i,j)} = \frac{2}{M(M-1)} \sum_{n=1}^{M} \sum_{m=1}^{M} \max\left(0, \theta_{\text{thre}} - \theta_{nm}\right), \tag{2}$$

$$\theta_{nm} = \cos^{-1} \left( \frac{(P_i^n)^{\text{T}} P_j^m}{\max(\|P_i^n\|_2, \epsilon) \cdot \max(\|P_j^m\|_2, \epsilon)} \right), \tag{3}$$

where $\theta_{nm}$ measures the angle between any two prompts, $n$ and $m$; $\epsilon$ is a scalar to avoid division by zero. Note that, $\mathcal{L}_{dd}^{(i,j)}$ encourages the angles between each prompt to be at least $\theta_{\text{thre}}$ degrees. Since $(i, j)$ is unordered Cartesian product of $\omega$, i.e. $(i, j) \in \{(i, j) \mid i \in \omega \wedge j \in \omega\}$, the inter-pool prompt discrepancy loss for the three pools can be expressed with $\mathcal{L}_{inter} = \mathcal{L}_{dd}^{(s,o)} + \mathcal{L}_{dd}^{(s,c)} + \mathcal{L}_{dd}^{(o,c)}$, and the intra-pool prompt diversity loss becomes $\mathcal{L}_{intra} = \mathcal{L}_{dd}^{(s,s)} + \mathcal{L}_{dd}^{(o,o)} + \mathcal{L}_{dd}^{(c,c)}$. As opposed to $\mathcal{L}_{inter}$, $\mathcal{L}_{intra}$ computes the angle between any two prompts within the same pool. Thus, it contains the case when $n = m$, for which we set $\theta_{\text{thre}} - \theta_{nm} = 0$.

## 4.2 Object-injected State Prompting

Akin to the query-key matching mechanism in other work [43, 42, 40], we utilize a fixed feature extractor $f(\cdot)$ to obtain a query feature $q(x) = f(x)[0, :]$, determining which prompts in the pool to be selected. However, pre-trained backbones are typically trained for object classification, thus under-performing for state representation learning. In addition, it is more difficult to predict the state classes due to their more abstract and fine-grained characteristics. To tackle this problem, we strategically inject object prompts to guide the selection of state prompts. Intuitively, once we have learned knowledge about the object class, it may be easier to predict the correct state class and avoid mistaken results. For instance, given an object is "heels", we can expect that the corresponding

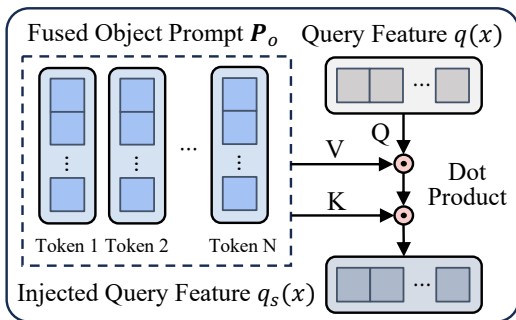

Figure 5: Architecture of object-injected state prompting. Query feature serves as Q, while fused object prompt serves as both K and V.

state is unlikely to be "canvas" or "plastic". To summarize, we select object and composition prompts in each pool based on the original query feature, which means $q_o(x) = q_c(x) = q(x)$; but for the selection of state prompts, we propose object-injected state prompting to ameliorate the query feature as shown in Fig. 5.

Specifically, we employ the fused object prompt $\boldsymbol{P}_o$ (see Sec. 4.3) to perform cross attention on the query feature $q(x)$, resulting in object-injected query feature $q_s(x)$ for the state prompt selection:

$$q_s(x) = \text{CrossAttn}(q(x), \boldsymbol{P}_o) = \text{Softmax}(\frac{q(x)W^Q \cdot \boldsymbol{P}_o W^K}{\sqrt{D}}) \cdot \boldsymbol{P}_o W^V, \tag{4}$$

where $W^Q$, $W^K$ and $W^V$ are learnable projections. To establish alignment between the query and the selected prompts, we optimize a surrogate loss for state, object and composition prompting jointly:

$$\mathcal{L}_{sur} = \sum_\omega \sum_{q_\omega} \text{COS}(f_\omega(x), K_\omega^{s_i}), \ \omega \in \{s, o, c\}, \tag{5}$$

where $\text{COS}(\cdot, \cdot)$ denotes cosine similarity, $\mathbf{K}_\omega$ represents the subset of top-k keys selected from $\mathbb{K}_\omega$, and $\{s_i\}_{i=1}^k$ is a subset of top-k indices from $[1, M]$ (prompt number). Despite the simplicity of the object-injected state prompting, it facilitates more judicious prompt selection within the state prompt pool, alleviating the hurdles posed by state learning.

## 4.3 Generalized-mean Prompt Fusion

After obtaining the selected top-k prompts $\{P_\omega^{s_i}\}_{i=1}^k$, the next step is fusing these prompts into a single prompt. It is general to utilize a simple mean pooling whereas it overlooks the relative importance of each prompt. Besides, when the prompts contain information that is unrelated or contradictory to current task, it is critical to strengthen useful prompts and eliminate irrelevant ones. To this end, we draw inspiration from generalized-mean pooling [29] and exploit generalized-mean (GeM) prompt fusion which is given by:

$$\boldsymbol{P}_\omega = \text{GeM}_\omega(P_\omega^{s_1}, P_\omega^{s_2}, \cdots, P_\omega^{s_k}) = \left(\frac{1}{k} \sum_{i=1}^k P_\omega^{s_i \eta}\right)^{\frac{1}{\eta}}, \omega \in \{s, o, c\}, \tag{6}$$

where $\eta$ is a learnable parameter. When $\eta = 1$, GeM becomes mean pooling; as $\eta$ approaches infinity ($\eta \to \infty$), it converges to max pooling. By taking over mean and max pooling, GeM learns to achieve an optimal fusion, mitigating the influence of irrelevant information present in the prompts.

## 4.4 Training and Inference

**Classification Objective.** We prepend three fused prompts (*i.e.* $\boldsymbol{P}_s$, $\boldsymbol{P}_o$ and $\boldsymbol{P}_c$) with $x_e$, which is the output from a ViT embedding layer $f_e(\cdot)$. The extended token sequence is $x_p = [\boldsymbol{P}_c; \boldsymbol{P}_s; \boldsymbol{P}_o; x_e]$.

Table 1: Avg Acc and FTT results on Split-Clothing (5 tasks) and Split-UT-Zappos (5 and 10 tasks). The best results are marked in **bold**. All results with standard deviations are averaged over three runs.

| Datasets | Split-Clothing (5 tasks) | | Split-UT-Zappos (5 tasks) | | Split-UT-Zappos (10 tasks) | |
|---|---|---|---|---|---|---|
| Metrics | Avg Acc($\uparrow$) | FTT($\downarrow$) | Avg Acc($\uparrow$) | FTT($\downarrow$) | Avg Acc($\uparrow$) | FTT($\downarrow$) |
| Upper Bound | 97.02±0.10 | - | 68.71±0.41 | - | 68.71±0.41 | - |
| EWC [10] | 47.89±0.87 | 52.75±0.44 | 37.59±2.06 | 55.70±2.76 | 24.63±0.94 | 61.31±2.29 |
| LwF [16] | 49.96±0.68 | 44.22±0.53 | 40.15±0.43 | 49.61±0.68 | 30.38±1.41 | 58.15±0.20 |
| iCaRL [32] | 68.65±0.41 | 31.74±1.89 | 37.78±2.14 | 55.06±3.50 | 31.40±1.96 | 59.65±2.40 |
| L2P [43] | 80.22±0.41 | 14.23±0.44 | 42.20±2.18 | 20.41±2.76 | 31.65±0.16 | 31.02±1.62 |
| Deep L2P++[43, 33] | 80.55±0.45 | 12.60±1.90 | 42.37±0.65 | 30.10±1.56 | 30.68±0.35 | 32.20±1.96 |
| Dual-Prompt [42] | 87.87±0.63 | 7.71±0.25 | 43.30±0.19 | 19.41±2.80 | 33.01±1.65 | 24.61±1.11 |
| CODA-Prompt [33] | 86.35±0.20 | 8.99±0.71 | 43.35±0.29 | 21.76±2.45 | 31.40±0.36 | 30.54±2.63 |
| LGCL [7] | 87.32±0.10 | 7.58±0.06 | - | - | 33.56±0.31 | **24.37**±0.56 |
| **Sim-CompILer** | 88.38±0.08 | 8.01±0.42 | 45.70±0.68 | 20.06±0.62 | 33.30±0.10 | 30.31±0.03 |
| **CompILer** | **89.21**±0.24 | **7.26**±0.60 | **46.48**±0.26 | **19.27**±0.75 | **34.43**±0.07 | 28.69±0.82 |

Then, we feed $x_p$ to a transformer encoder layer $f_r(\cdot)$ and achieve $\boldsymbol{P}_s^r$, $\boldsymbol{P}_o^r$ and $\boldsymbol{P}_c^r$ for classifying state, object and composition classes, respectively. We estimate the probability via a classifier $\varphi_\omega(\cdot)$: $p(\omega \mid x) = \varphi_\omega(\boldsymbol{P}_\omega^r)$. For each image $x$, we denote its ground-truth distribution over labels with $q(\omega \mid x)$. When $\omega$ is consistent with the ground truth, then $q(\omega \mid x) = 1$; otherwise, $q(\omega \mid x) = 0$. As a result, the cross entropy (CE) loss used for classification objective is:

$$\mathcal{L}_{CE}^\omega = -\sum_{\omega=1}^{\Omega} q(\omega \mid x) \log p(\omega \mid x), \Omega \in [|\mathcal{S}|, |\mathcal{O}|, |\mathcal{C}|], \tag{7}$$

where $\Omega$ represents the number of classes. However, the model optimized with a standard CE loss is easily affected by noisy samples during training. Instead, we advocate using a symmetric cross entropy loss (SCE) [41], which incorporates an additional term called reverse cross entropy (RCE), to mitigate the impact of noisy data. Contrary to CE, the formula for RCE loss is defined as:

$$\mathcal{L}_{RCE}^\omega = -\sum_{\omega=1}^{\Omega} p(\omega \mid x) \log q(\omega \mid x), \omega \in \{s, o, c\}. \tag{8}$$

Then, the SCE loss combines two loss terms by $\mathcal{L}_{SCE}^\omega = \mathcal{L}_{CE}^\omega + \alpha \mathcal{L}_{RCE}^\omega$, where $\alpha$ is a hyper-parameter that controls the weight of the RCE term. As a result, the whole SCE loss becomes $\mathcal{L}_{SCE} = \mathcal{L}_{SCE}^c + \beta(\mathcal{L}_{SCE}^s + \mathcal{L}_{SCE}^o)$, where $\beta$ adjusts the weights between primitives and compositions.

**Total Loss.** The total loss for training the whole CompILer model is:

$$\mathcal{L}_{total} = \lambda_1 \mathcal{L}_{inter} + \lambda_2 \mathcal{L}_{intra} + \lambda_3 \mathcal{L}_{sur} + \mathcal{L}_{SCE}, \tag{9}$$

where $\lambda_1$, $\lambda_2$, $\lambda_3$ are hyper-parameters balancing different terms.

**Inference.** During inference, we incorporate the primitive probabilities to aid the composition probability. Hence, the final probability for composition classification is expressed with:

$$p_{\text{final}}(c \mid x) = p(c \mid x) + \mu(p(s \mid x) + p(o \mid x)), \tag{10}$$

where $\mu$ adjusts the probabilities.

## 5 Experiments

### 5.1 Datasets and Metrics

We conduct experiments on two newly split datasets: Split-Clothing and Split-UT-Zappos as elucidated in Section 3.2. We assess the overall performance on compositions using Average Accuracy (**Avg Acc**) and Forgetting (**FTT**). A higher Avg Acc signifies stronger recognition abilities, while a lower FTT indicates improved resilience against forgetting. Additionally, we provide individual Average Accuracy scores on states and objects, denoted as **State** and **Object** for simplicity. These metrics imply the ability to recognize fine-grained primitives. Furthermore, we calculate the Harmonic Mean (**HM**) between State and Object, *i.e.* $HM = 2 \times \frac{(State \times Object)}{(State + Object)}$. We provide more emphasis to Avg Acc and HM due to their more comprehensive assessment. Avg Acc encompasses the plasticity and stability [33, 2] and HM provides a holistic evaluation on both state and object.

## 5.2 Implementation Details

For a fair comparison with previous works [43, 42, 2, 33], we also employ ViT B/16 [1] pretrained on the ImageNet 1K dataset as the feature extractor and backbone. For multi-pool prompt learning, the size of each pool is set to 20, and each prompt has 5 tokens. We select top-5 prompts from each pool and generate a fused prompt. During training, we utilize the Adam optimizer [9] with a batch size of 16. The whole CompILer undergoes training for 25 epochs on the Split-Clothing, for 10 epochs on the 5-task Split-UT-Zappos, and for 3 epochs on

Table 2: State, Object and HM results on Split-Clothing. The best results are marked in **bold**.

| Datasets | Split-Clothing (5 tasks) | | |
|---|---|---|---|
| Metrics | State | Object | HM |
| Upper Bound | 97.44±0.08 | 97.09±0.10 | 97.26±0.08 |
| EWC [10] | 86.49±0.97 | 52.72±1.30 | 675.50±0.97 |
| LwF [16] | 87.11±0.66 | 54.57±0.69 | 67.10±0.33 |
| iCaRL [32] | 91.21±1.05 | 71.70±0.99 | 80.28±0.74 |
| L2P [43] | 83.03±0.42 | 95.56±0.57 | 88.85±0.16 |
| Dual-Prompt [42] | 90.77±0.25 | 94.18±0.31 | 92.45±0.20 |
| LGCL [7] | 91.45±0.20 | 94.87±0.33 | 93.13±0.10 |
| **Sim-CompILer** | 91.15±0.10 | 96.32±0.02 | 93.66±0.02 |
| **CompILer** | **91.81**±0.23 | **96.67**±0.01 | **94.18**±0.06 |

the 10-task Split-UT-Zappos. For the Split-Clothing and the 10-task Split-UT-Zappos, we set the learning rate to 0.03, while we use a learning rate of 0.02 for the 5-task Split-UT-Zappos. Note that, for all the methods, their results are averaged over **three runs** with the corresponding standard deviations reported to mitigate the influence of random factors.

As there are a few hyper-parameters in the model, we conduct a rigorous tuning on them. For instance, we set $\theta_{thre}$ to $\frac{\pi}{2}$ for all settings. For Split-Clothing, the loss weights $\lambda_1$ and $\lambda_3$ are set to 0.1; $\lambda_2$ is set to $10^{-7}$; $\alpha$ and $\beta$ for SCE loss are 0.006 and 0.3, and the parameter $\mu$ during inference is 0.5. For 5-task Split-UT-Zappos, $\lambda_1$, $\lambda_2$, $\lambda_3$, $\alpha$, $\beta$ and $\mu$ are set to 1.0, $3 \times 10^{-6}$, 0.7, 0.01, 0.7 and 0.02, respectively. For 10-task Split-UT-Zappos, $\lambda_1$, $\lambda_2$, $\lambda_3$, $\alpha$, $\beta$ and $\mu$ are set to 0.5, $10^{-7}$, 0.1, 0.05, 0.4 and 0.03. We elaborate more details on hyper-parameter analysis in the appendix.

## 5.3 Compared Baselines

To demonstrate the effectiveness of the proposed method, we compare CompILer with state-of-the-art incremental learning methods, including prompt-free approaches [10, 16, 32] and prompt-based methods [43, 42, 33, 7]. All the methods are rehearsal-free except iCaRL [32]. Note that, due to LGCL [7] relying on CLIP [30] to achieve language guidance at the task level, it is limited by the length of class names per task. Thereby, LGCL fails to operate on the 5-task Split-UT-Zappos since the total length of class names exceeds the limitation.

To streamline our CompILer, we further implement a simplified version named **Sim-CompILer** and report its results. Sim-CompILer is optimized using cross entropy loss and is comprised solely of multi-pool prompt learning and generalized-mean prompt fusion. In other words, we exclude the object-injected prompting, directional decoupled loss, and reverse cross entropy loss, resulting in a large reduction of hyperparameters to only $\beta$, $\lambda_3$, and $\mu$.

## 5.4 Comparison with the State-of-the-arts

The compared results on Avg Acc and FTT are reported in Table 1. Overall, CompILer consistently outperforms all competitors on Avg Acc by a significant margin. For FTT scores, CompILer excels previous methods with 0.32% on the 5-task Split-Clothing and with 0.14% on the 5-task Split-UT-Zappos, while falling behind Dual-Prompt [42] and LGCL [7] for the 10-task Split-UT-Zappos. We notice that, the main reason is these methods sacrifice more plasticity for lower forgetting rates. Besides, the number of model parameters in these methods dynamically increases along with more incremental tasks arriving, whereas our CompILer does not rely on imposing task-specific parameters to reduce the forgetting.

We also report the primitives accuracy and their HM in Table 2 and Table 3. Likewise, our method surpasses other methods considerably in terms of State and HM. Interestingly, the prompt-free methods [16, 10, 32] achieve higher accuracy in state prediction than object prediction for Split-Clothing, which is contrary to other results. This is because the states in Split-Clothing are color-related descriptions, which are easier to capture with the help of parameter fine-tuning. The prompt-based methods do not exhibit this phenomenon because their pre-trained backbones are initially

Table 3: State, Object and HM results on Split-UT-Zappos (5 tasks) and Split-UT-Zappos (10 tasks).

| Datasets | Split-UT-Zappos (5 tasks) | | | Split-UT-Zappos (10 tasks) | | |
|---|---|---|---|---|---|---|
| Metrics | State | Object | HM | State | Object | HM |
| Upper Bound | 75.10±0.10 | 88.13±0.03 | 81.90±0.06 | 75.10±0.10 | 88.13±0.03 | 81.90±0.06 |
| EWC [10] | 47.95±1.26 | 76.53±0.91 | 58.90±0.53 | 39.29±2.69 | 67.64±1.97 | 49.69±2.30 |
| LwF [16] | 53.13±1.08 | 75.48±0.82 | 62.35±0.31 | 38.70±2.33 | 68.90±1.97 | 49.54±1.30 |
| iCaRL [32] | 51.71±0.95 | 75.03±0.49 | 61.22±0.78 | 38.94±2.01 | 67.10±1.05 | 49.27±1.58 |
| L2P [43] | 52.20±2.92 | 79.05±0.01 | 62.87±1.61 | 42.66±0.87 | 76.60±0.03 | 54.80±0.55 |
| Dual-Prompt [42] | 52.25±0.77 | 77.46±0.05 | 62.40±0.34 | 44.34±1.61 | 77.92±0.37 | 56.51±1.11 |
| LGCL [7] | - | - | - | 43.44±0.79 | **78.64**±0.64 | 55.96±0.43 |
| **Sim-CompILer** | 55.93±1.23 | **79.69**±0.06 | 65.72±0.53 | 45.88±0.38 | 75.72±0.67 | 57.14±0.06 |
| **CompILer** | **56.85**±0.34 | 79.56±0.04 | **66.31**±0.15 | **46.27**±1.56 | 76.65±1.19 | **57.69**±0.42 |

Table 5: Ablative experiments for (a) object-injected state prompting, (b) prompt fusion method.

(a) Object-injected state prompting.

| Dataset | Split-Clothing (5 tasks) | | |
|---|---|---|---|
| Metrics | Avg Acc | FTT($\downarrow$) | HM |
| None | 88.45±0.10 | 7.93±0.11 | 93.70±0.03 |
| S→O | 88.27±0.02 | 7.99±0.05 | 93.67±0.01 |
| O→S | **89.21**±0.24 | **7.26**±0.60 | **94.18**±0.06 |

(b) Prompt fusion method.

| Dataset | Split-Clothing (5 tasks) | | |
|---|---|---|---|
| Metrics | Avg Acc | FTT($\downarrow$) | HM |
| Max | 84.70±0.64 | 12.24±2.25 | 91.54±0.30 |
| Mean | 87.80±0.12 | 7.82±0.01 | 93.38±0.03 |
| GeM | **89.21**±0.24 | **7.26**±0.60 | **94.18**±0.06 |

trained for object classification, and are frozen across incremental sessions. As the performance improvements are mainly attributed to the accuracy of state recognition, it suggests that our model enhances the understanding on fine-grained compositionality.

## 5.5 Ablation Study and Analysis

**Effect of multi-pool prompt learning.** This experiment aims to delineate the contribution of three pools in CompILer. We firstly implement a baseline model with composition prompt pool only. Building upon the baseline, we develop two additional models, which incorporate either object or state prompt pool. As reported in Table 4, the inclusion of primitive prompt pool yields consistent gains over the baseline. Furthermore, the best results are achieved when the model integrates all three pools simultaneously. This experiment signifies the significant necessity of exploiting multiple prompt pools for composition-IL.

Table 4: Ablation study on multi-pool prompt learning with Split-Clothing dataset.

| Prompt Pool | | | Split-Clothing (5 tasks) | | |
|---|---|---|---|---|---|
| C | S | O | Avg Acc | FTT($\downarrow$) | HM |
| ✓ | | | 80.22±0.41 | 14.23±0.44 | 88.85±0.16 |
| ✓ | | ✓ | 88.10±0.11 | 7.79±0.04 | 93.55±0.04 |
| ✓ | ✓ | | 88.09±0.50 | **7.26**±0.54 | 93.52±0.13 |
| ✓ | ✓ | ✓ | **88.38**±0.08 | 8.01±0.42 | **93.66**±0.02 |

**Effect of object-injected state prompting.** To provide insights into object-injected state prompting, we compare three models: None (vanilla model), S→O (state-injected object prompting) and O→S (object-injected state prompting). As shown in Table 5a, compared to the None model, the S→O exhibits a decrease in all metrics, implying that state prompts may interfere with the selection of object prompts. On the contrary, O→S outperforms the None model as we expect. This phenomenon validates our motivation that state recognition is harder than object recognition, and thereby the former cannot help the latter easily. Yet, it is a promising direction for future research.

**Effect of generalized-mean prompt fusion.** This study aims to study the impact of prompt fusion on CompILer. As shown in Table 5b, GeM performs better than both max and mean pooling across various metrics. It validates the benefit of GeM on mitigating irrelevant information in the selected prompts. as it may hamper the model's attention on image tokens.

**Effect of loss functions.** As shown in Table 6, we investigate the influence of loss functions used in our model, including directional decoupled loss ($\mathcal{L}_{inter}$ and $\mathcal{L}_{intra}$) and symmetric cross entropy loss ($\mathcal{L}_{CE}$ and $\mathcal{L}_{RCE}$). The baseline model (the first row) includes all modules but is trained by cross entropy loss only. By adding the RCE loss, the model is equivalent to training with the SCE loss, which help to improve the robustness to noisy labels. The use of either $\mathcal{L}_{inter}$ or $\mathcal{L}_{intra}$ improves the performance on both datasets, and synchronously applying them witnesses all-around improvements

Table 6: Ablate the loss functions on Split-Clothing and Split-UT-Zappos.

| Loss function | | | | Split-Clothing (5 tasks) | | Split-UT-Zappos (5 tasks) | |
|---|---|---|---|---|---|---|---|
| $\mathcal{L}_{CE}$ | $\mathcal{L}_{RCE}$ | $\mathcal{L}_{inter}$ | $\mathcal{L}_{intra}$ | Avg Acc | FTT($\downarrow$) | Avg Acc | FTT($\downarrow$) |
| ✓ | | | | 88.17±0.08 | 8.08±0.27 | 44.83±0.15 | 19.49±2.93 |
| ✓ | ✓ | | | 88.36±0.37 | 8.33±0.11 | 45.47±0.07 | 20.14±0.43 |
| ✓ | | ✓ | | 88.32±0.56 | 7.82±0.64 | 45.58±0.04 | 19.64±0.37 |
| ✓ | | | ✓ | 88.42±0.30 | 8.23±0.06 | 45.62±0.13 | 20.13±0.14 |
| ✓ | | ✓ | ✓ | 88.61±0.61 | 7.72±0.87 | 46.01±0.69 | 19.50±0.86 |
| ✓ | ✓ | ✓ | ✓ | **89.21**±0.24 | **7.26**±0.60 | **46.48**±0.26 | **19.27**±0.75 |

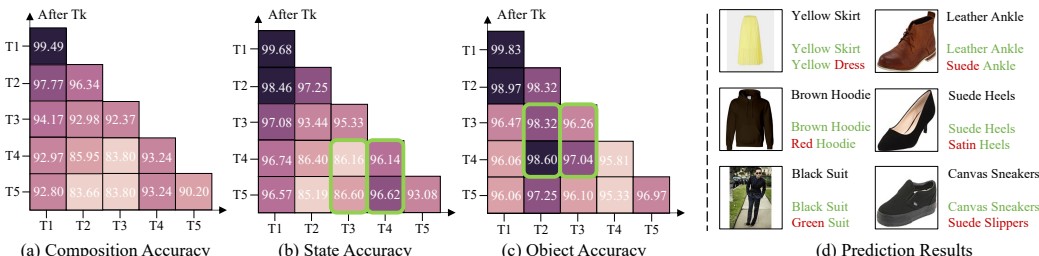

Figure 6: Results and analysis. (a) to (c) show accuracy of CompILer on composition, state, and object for each task in Split-Clothing. The x-axis represents the test stream, and the y-axis denotes the status after training the $T_k$ task. Darker background color indicates higher accuracy. (d) displays some images and their predictions: top row is GT, middle row is CompILer prediction, and bottom row is L2P [43] prediction. Green indicates correct predictions, while red indicates incorrect predictions.

compared to the baseline. Eventually, we achieve the best results when combing all the loss terms during training.

## 5.6 Additional Results and Analysis

In order to study the repeatability characteristic in composition-IL, we exhibit more results on Split-Clothing in Fig. 6: in (a), it shows a decreasing trend in composition accuracy along with the introduction of new tasks; however, the green rectangles in (b) and (c) showcase that the accuracy occasionally increases as more tasks are learned. We conjecture the reason is mostly attributed to the re-occurrence of primitive concepts. This forward transfer is critical for incremental learners. We compare the composition predictions between CompILer and L2P [43] in Fig. 6 (d). CompILer predicts all the images correctly, while L2P makes some mistakes, particularly for state labels. This limitation arises from an excessive focus on the dominant object primitive, while weakening the attention toward state primitive. Fortunately, CompILer relieves the bias toward object classes, and enhances the perception on state classes.

## 6 Conclusion

In this paper, we have proposed a novel task coined compositional incremental learning (compostion-IL), which is stumbled by ambiguous composition boundary. To tackle it, we develop a learning-to-prompt model, namely CompILer. Our model exploits multi-pool prompt learning to model composition and primitive concepts, object-injected state prompting to improve the selection of state prompts, and generalized-mean prompt fusion to eliminate irrelevant information. Extensive experiments on two tailored datasets show that CompILer achieves state-of-the-art performance. In the future, it is challenging yet potential to consider reasoning multiple state classes per object.

## 7 Acknowledgments

This work was supported in part by the National Natural Science Foundation of China under Grant Numbers 62102061, 62272083 and 62472066, and in part by the Open Projects Program of State Key Laboratory of Multimodal Artificial Intelligence Systems.

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

# A  Appendix

In addition to the content in the main paper, this appendix elaborates more details on the datasets, algorithm procedure, empirical analysis, hyper-parameter analysis, and quantitative and qualitative experiments. Meanwhile, please refer to the ***source code*** and some ***data samples*** included in the supplementary material we submit.

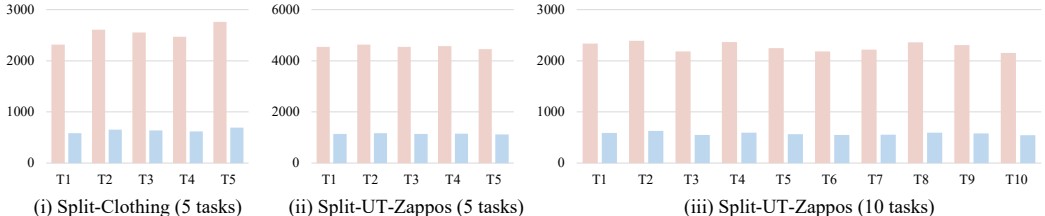

Figure 7: Number of images per task. The horizontal axis indicates the task ID, while the vertical axis represents the corresponding number of images. The colors pink and blue denote training and testing datasets, respectively. Overall, the data distributions are relatively balanced across tasks.

## A.1  More on Dataset Details

We provide more information about two newly tailored datasets (Split-Clothing and Split-UT-Zappos) and extensively discuss the limitations of Split-UT-Zappos. The number of images in each task is shown in Fig. 7.

**Split-Clothing** is derived from Clothing16K, which is originally used for multi-label classification tasks and consists of 37 state-object compositions and 16,170 images scraped from Google, Bing, and DuckDuckGo. We sorted these compositions by the number of images and selected the top 35 compositions, resulting in a total of 15,915 images in the Split-Clothing dataset. Specifically, the object in Split-Clothing includes various types of garments such as "dresses" and "hoodies", while the state delineates the color of the clothing, such as "silver" and "yellow". We allocate 80% of the dataset for training and the remaining 20% for testing. We split this dataset for 5 incremental tasks.

**Split-UT-Zappos** is a subset of UT-Zappos50K, which is a large shoe dataset consisting of 50,025 catalog images collected from Zappos.com. This dataset is originally created for an online shopping task and we selected a total of 28,497 images to comprise the Split-UT-Zappos. The state classes in Split-UT-Zappos describe different materials (*e.g.* canvas and leather) and the object classes are related to footwear (*e.g.* heels and slippers). Consistent with Split-Clothing, we employ 80% of the images for training and 20% for testing. We split this dataset for either 5 or 10 incremental tasks.

**Limitations of Split-UT-Zappos.** It is noteworthy that, even for the upper bound model (*i.e.* supervised learning with all the data), satisfactory performance remains elusive on Split-UT-Zappos as shown in Table 1 and Table 3. This deficiency can be attributed to the pronounced long-tail distributions and invisible states inherent in UT-Zappos50K. For example, the composition "Faux Fur Slippers" comprises a mere 25 training images, whereas "Leather Sandals" accounts for 1783. This disproportionate distribution predisposes the severe bias towards the compositions prevalent in the head of the distributions, while inadequately capturing those in the tail. Besides, some states in Split-UT-Zappos lik,e "Leather" vs "Synthetic Leather", are material differences that are not always visible as visual transformations. The deficient state description poses a greater challenge for classification, thereby resulting in suboptimal upper bound results. We contend that tackling the long-tail distribution challenge within Split-UT-Zappos and constructing more balanced datasets conducive to composition-IL represent promising avenues for future research.

## A.2  More on Training Procedure

We summarize the training procedure of the proposed CompILer in Algorithm 1.

**Algorithm 1:** Training Procedure of CompILer for composition-IL

---

1 **Input:** Training data for $T$ tasks, where each image $x$ has a set of three labels: $c$, $s$ and $o$. Three prompt pools $\mathbb{P}_\omega$ with corresponding keys $\mathbb{K}_\omega$, where $\omega \in \{s, o, c\}$. Pre-trianed feature extractor $f(\cdot)$, pre-trained input embedding layer $f_e(\cdot)$, pre-trained transformer encoder layer $f_r(\cdot)$, cross attention layer CrossAttn$(\cdot, \cdot)$, classifiers $\varphi_\omega(\cdot)$, GeM fusion GeM$_\omega(\cdots)$.

2 **Initialize:** $\mathbb{P}_\omega$, $\mathbb{K}_\omega$, CrossAttn$(\cdot, \cdot)$, $\varphi_\omega(\cdot)$, GeM$_\omega(\cdots)$

3 **for** $t = 1, \cdots, T$ **do**

4    **for** *each sample in the batch* **do**

5       Estimate the prompt-specific loss $\mathcal{L}_p = \lambda_1 \mathcal{L}_{inter} + \lambda_2 \mathcal{L}_{intra}$;

6       Calculate query feature of object and composition $q_o(x) = q_c(x) = f(x)[0, :]$;

7       Lookup top-k object keys $\mathbf{K}_o$ and composition keys $\mathbf{K}_c$;

8       Select top-k prompts associated with the keys in $\mathbb{P}_o$ and $\mathbb{P}_c$;

9       Fuse the selected prompts by $\boldsymbol{P}_o = \text{GeM}_o(P_o^{s_1}, P_o^{s_2}, \cdots, P_o^{s_k})$ and $\boldsymbol{P}_c = \text{GeM}_c(P_c^{s_1}, P_c^{s_2}, \cdots, P_c^{s_k})$;

10       Perform object-injected state prompting by: $q_s(x) = \text{CrossAttn}(f(x)[0, :], \boldsymbol{P}_o)$;

11       Lookup top-k state keys $\mathbf{K}_s$;

12       Select top-k state prompts associated with the keys in $\mathbb{P}_s$;

13       Fuse the selected state prompts by $\boldsymbol{P}_s = \text{GeM}_s(P_s^{s_1}, P_s^{s_2}, \cdots, P_s^{s_k})$;

14       Calculate the input embedding sequence $x_e = f_e(x)$;

15       Prepending $x_e$ with fused prompts by $x_p = [\boldsymbol{P}_c; \boldsymbol{P}_s; \boldsymbol{P}_o; x_e]$;

16       Feed $x_p$ to a transformer encoder layer $f_r(\cdot)$ and achieve $\boldsymbol{P}_s^r$, $\boldsymbol{P}_o^r$ and $\boldsymbol{P}_c^r$

17       Calculate the probability by $p(\omega \mid x) = \varphi_\omega(\boldsymbol{P}_\omega^r)$;

18       Estimate per sample loss $\mathcal{L}_x = \mathcal{L}_{SCE} + \lambda_3 \mathcal{L}_{sur}$;

19    **end**

20    Calculate per batch loss $\mathcal{L}_B$ by accumulating $(\mathcal{L}_x + \mathcal{L}_p)$.

21 **end**

22 **Output:** The network and its network parameters $\Phi$.

---

### A.3 Empirical Analysis on Learned Prompts

In Section 4.1, we design multi-pool prompt learning to learn visual representations of state, object, and composition, respectively. By applying inter-pool discrepant loss and intra-pool diversified loss, we ensure discrepancy across pools and diversity within each pool. To validate the effectiveness of the approach, we conduct *t*-SNE visualization on the prompt pools within the three experiment settings. Figure 8 illustrates that the learned knowledge across the three pools is remarkably discriminative, suggesting that each pool has effectively captured unique features. Meanwhile, the prompts within each pool exhibit a relatively wide dispersion demonstrating that the learned information within each pool is diversified.

### A.4 Empirical Analysis of Generalized-mean Prompt Fusion

The detailed operation of Generalized-mean Prompt Fusion is elaborated on Section 4.3. The parameter $\eta$ in Eq. 6 can be learnable as this operation is differentiable, allowing it to be included in the whole back-propagation process. To prove that, the corresponding derivatives of Eq. 6 are given by:

$$
\begin{aligned}
\frac{\partial \boldsymbol{P}_\omega}{\partial P_\omega^{s_i}} &= \frac{1}{k} \boldsymbol{P}_\omega^{1-\eta} P_\omega^{s_i \eta - 1}, \\
\frac{\partial \boldsymbol{P}_\omega}{\partial \eta} &= \frac{\boldsymbol{P}_\omega}{\eta^2} \left( \log \frac{k}{\sum_{i=1}^k P_\omega^{s_i \eta}} + \eta \frac{\sum_{i=1}^k P_\omega^{s_i \eta} \log P_\omega^{s_i}}{\sum_{i=1}^k P_\omega^{s_i \eta}} \right).
\end{aligned}
\tag{11}
$$

### A.5 Empirical Analysis of Symmetric Cross Entropy Loss

In this section, we delve into why symmetric cross entropy (SCE) loss can effectively mitigate noisy data in the dataset. We define two distributions, $q$ and $p$, where $q = (k \mid x)$ represents the ground truth class distribution for sample $x$ and $p = (k \mid x)$ represents the predicted class distribution. The

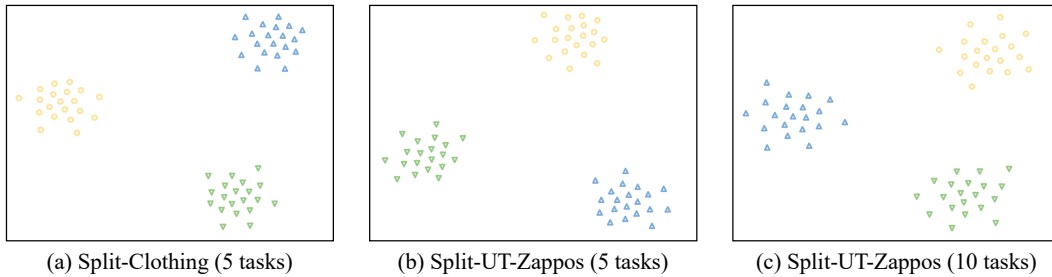

| (a) Split-Clothing (5 tasks) | (b) Split-UT-Zappos (5 tasks) | (c) Split-UT-Zappos (10 tasks) |

Figure 8: The *t*-SNE visualization of learned prompts on Split-Clothing (5 tasks), Split-UT-Zappos(5 tasks), and Split-UT-Zappos (10 tasks). The composition prompts are colored in yellow, the state prompts in green, and the object prompts in blue.

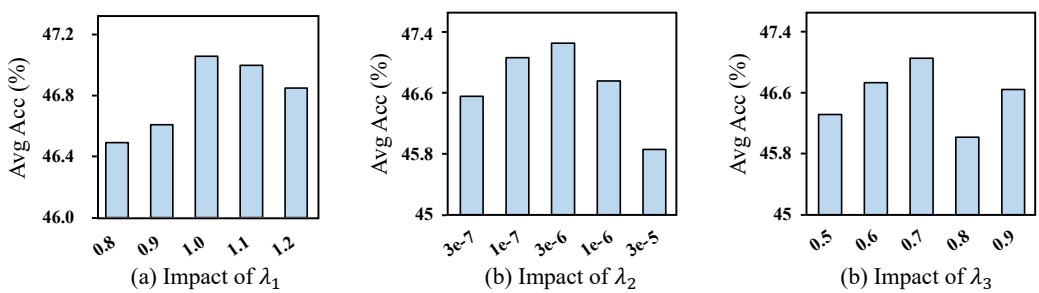

| (a) Impact of $\lambda_1$ | (b) Impact of $\lambda_2$ | (b) Impact of $\lambda_3$ |

Figure 9: Impact of hyper-parameters on average accuracy in Split-UT-Zappos (5 tasks).

relation between the cross entropy $H(q, p)$ and KL-divergence $KL(q \parallel p)$ is expressed as:

$$KL(q \parallel p) = H(q, p) - H(q), \tag{12}$$

in which $H(q)$ denotes the entropy of $q$. From the perspective of KL divergence, the optimization objective is to make the prediction distribution $p$ closely resemble the ground truth distribution $q$, essentially minimizing their KL divergence. However, in the presence of noise within the dataset, $q$ may not accurately reflect the ground-truth class distribution, whereas $p$ might reveal the real distribution. Therefore, it becomes necessary to consider the reverse direction of KL divergence, *i.e.* $KL(p \parallel q)$ and construct symmetric KL divergence represented by:

$$SKL(q \parallel p) = KL(q \parallel p) + KL(p \parallel q). \tag{13}$$

By transferring this idea from KL divergence to cross entropy, the formula for symmetric cross entropy is given by:

$$SCE = CE + RCE = H(q, p) + H(p, q), \tag{14}$$

where $RCE = H(p, q)$ is the reverse counterpart of $H(q, p)$, known as reverse cross entropy.

### A.6 More on Hyper-parameter Analysis

We investigate the impact of several hyper-parameters, as depicted in Fig. 9. Specifically, we explore the influence of weights in the total loss. We present the results of tuning the weight $\lambda_1$, $\lambda_2$, and $\lambda_3$ on the Split-UT-Zappos (5 tasks). The average accuracy shown in Fig. 9(a), (b), and (c) initially exhibits an increasing trend, followed by an overall decreasing trend. CompILer reaches its peak performance when $\lambda_1 = 1.0$, $\lambda_2 = 3e - 6$, and $\lambda_3 = 0.7$.

### A.7 More on Ablation Studies

**Effect of multi-pool prompt learning.** In addition to presenting the results of Avg ACC, FTT, and HM on Split-Clothing (5 tasks) for multi-pool prompt learning, we also include the remaining results for State and Object in Table 7. An intriguing observation is that the model achieves the best

Table 7: Ablate the multi-pool prompt learning on Split-Clothing (5 tasks). C, S, and O denote the composition pool, state pool, and object pool, respectively.

| Prompt Pool | | | Split-Clothing (5 tasks) | | | | |
|---|---|---|---|---|---|---|---|
| C | S | O | Avg Acc | FTT(↓) | State | Object | HM |
| ✓ | | | 80.01 | 12.85 | 83.25 | 94.45 | 88.50 |
| ✓ | | ✓ | 88.10±0.11 | 7.79±0.04 | 90.42±0.02 | **96.91**±0.10 | 93.55±0.04 |
| ✓ | ✓ | | 88.09±0.50 | **7.26**±0.54 | **91.67**±0.19 | 96.00±0.03 | 93.52±0.13 |
| ✓ | ✓ | ✓ | **88.38**±0.08 | 8.01±0.42 | 91.15±0.10 | 96.32±0.02 | **93.66**±0.02 |

Table 8: Ablate the guidance on Split-Clothing (5 tasks).

| Dataset | Split-Clothing (5 tasks) | | | | |
|---|---|---|---|---|---|
| Guidance | Avg Acc | FTT(↓) | State | Object | HM |
| None | 88.45±0.10 | 7.93±0.11 | 91.26±0.11 | 96.28±0.01 | 93.70±0.03 |
| S→O | 88.27±0.02 | 7.99±0.05 | 90.99±0.22 | 96.53±0.19 | 93.67±0.01 |
| O→S | **89.21**±0.24 | **7.26**±0.60 | **91.81**±0.23 | **96.67**±0.01 | **94.18**±0.06 |

results for Object when the object pool is used exclusively, while it reaches the peak performance for State when only the state pool is introduced. This phenomenon suggests that incorporating the primitive prompt pools significantly aids in learning the representations of the corresponding primitives. Furthermore, when the model integrates the state prompt pool alone, not only does the accuracy of state is improved, but the accuracy of object also experiences significant gains, and vice versa. This implies that the learning of one type of primitive representation is crucial for the learning of another. Only when all three pools are integrated does the model achieve state-of-the-art performance in the most important metrics, namely average accuracy and HM.

**Effect of object-injected state prompting.** In this section, we report the remaining results for object-injected state prompting on Split-Clothing (5 tasks) as shown in Table 8. State-injected object prompting (S→O) introduces interference information for composition learning, resulting in a decline across all metrics expect Object. By contrast, object-injected state prompting (O→S) leads to a significant improvement across all metrics, demonstrating the effectiveness of the proposed method.

**Effect of generalized-mean prompt fusion.** We show the intact results of the ablation study on pooling methods on Split-Clothing (5 tasks) in Fig. 9. It is obvious that max pooling hits the lowest result, while integrating mean pooling leads to a stable growth. Only deploying Generalized-mean pooling achieves the state-of-the-art results, surging from 84.70 to 89.21 in Avg Acc and plunging to 7.26 in FTT. Consequently, the theory that GeM pooling can diminish the irrelevant information and emphasize the crucial aspects is proved.

**Effect of loss functions.** Eventually, we present supplementary ablation experiments on loss functions in Table 10 and 11. The first row indicates that the model is optimized using cross entropy loss exclusively which achieves the poorest results. Furthermore, integrating either reverse cross entropy loss, inter-pool prompt discrepancy loss, or intra-pool diversity loss contributes to improvements across all metrics. Notably, the effectiveness of reverse cross entropy loss is higher on Split-UT-Zappos (5 tasks) compared to that on Split-Clothing (5 tasks). To conclude, the best performance is achieved when all loss functions are simultaneously employed.

## A.8   More on Qualitative Results

More qualitative results are presented to demonstrate the effectiveness of CompILer. We conducted a comparative analysis of the prediction results between CompILer and L2P [43] on Split-Clothing (5 tasks), Split-UT-Zappos (5 tasks) and Split-UT-Zappos (10 tasks). The ground truth is presented in the first row, followed by the prediction results of CompILer and L2P in the second and third rows, respectively. As illustrated in the Fig. 10, CompILer achieves correct predictions for all the images, while L2P consistently yields wrong classification results due to ambiguous composition boundary. Specifically, L2P exhibits a higher occurrence of misclassification on states compared to objects, which can be attributed to the disproportionate emphasis placed on objects, diminishing the model's focus on states. This experiment proves that CompILer effectively enhances the model's attention towards states, resulting in an improved overall fine-grained perception capability.

Table 9: Ablate the pooling on Split-Clothing (5 tasks).

| Dataset | Split-Clothing (5 tasks) | | | | |
|---|---|---|---|---|---|
| Pooling | Avg Acc | FTT($\downarrow$) | State | Object | HM |
| Max | 84.70±0.64 | 12.24±2.25 | 86.79±0.96 | 96.84±0.01 | 91.54±0.30 |
| Mean | 87.80±0.12 | 7.82±0.01 | 90.78±0.12 | 96.14±0.01 | 93.38±0.03 |
| GeM | **89.21**±0.24 | **7.26**±0.60 | **91.81**±0.23 | **96.67**±0.01 | **94.18**±0.06 |

Table 10: Ablate the loss function on Split-Clothing (5 tasks).

| Loss function | | | | Split-Clothing (5 tasks) | | | | |
|---|---|---|---|---|---|---|---|---|
| $\mathcal{L}_{CE}$ | $\mathcal{L}_{RCE}$ | $\mathcal{L}_{inter}$ | $\mathcal{L}_{intra}$ | Avg Acc | FTT($\downarrow$) | State | Object | HM |
| ✓ | | | | 88.17±0.08 | 8.08±0.27 | 90.99±0.21 | 96.41±0.08 | 93.62±0.03 |
| ✓ | ✓ | | | 88.36±0.37 | 8.33±0.11 | 90.88±0.30 | 96.64±0.05 | 93.67±0.06 |
| ✓ | | ✓ | | 88.32±0.56 | 7.82±0.64 | 90.85±0.57 | 96.61±0.07 | 93.66±0.12 |
| ✓ | | | ✓ | 88.42±0.30 | 8.23±0.06 | 91.18±0.04 | 96.44±0.10 | 93.73±0.06 |
| ✓ | | ✓ | ✓ | 88.61±0.61 | 7.72±0.87 | 90.94±0.68 | 96.85±0.02 | 93.81±0.17 |
| ✓ | ✓ | ✓ | ✓ | **89.21**±0.24 | **7.26**±0.60 | **91.81**±0.23 | **96.67**±0.01 | **94.18**±0.06 |

Table 11: Ablate the loss function on Split-UT-Zappos (5 tasks).

| Loss function | | | | Split-UT-Zappos (5 tasks) | | | | |
|---|---|---|---|---|---|---|---|---|
| $\mathcal{L}_{CE}$ | $\mathcal{L}_{RCE}$ | $\mathcal{L}_{inter}$ | $\mathcal{L}_{intra}$ | Avg Acc | FTT($\downarrow$) | State | Object | HM |
| ✓ | | | | 44.83±0.15 | 19.49±2.93 | 55.07±0.25 | 79.06±0.06 | 64.92±0.18 |
| ✓ | ✓ | | | 45.47±0.07 | 20.14±0.43 | 55.92±0.05 | 79.14±0.13 | 65.47±0.03 |
| ✓ | | ✓ | | 45.58±0.04 | 19.64±0.37 | 56.02±0.04 | 79.25±0.01 | 65.64±0.01 |
| ✓ | | | ✓ | 45.62±0.13 | 20.13±0.14 | 55.98±0.24 | 79.45±0.20 | 65.68±0.08 |
| ✓ | | ✓ | ✓ | 46.01±0.69 | 19.50±0.86 | 56.31±0.72 | 79.53±0.05 | 65.94±0.40 |
| ✓ | ✓ | ✓ | ✓ | **46.48**±0.26 | **19.27**±0.75 | **56.85**±0.34 | **79.56**±0.04 | **66.31**±0.15 |

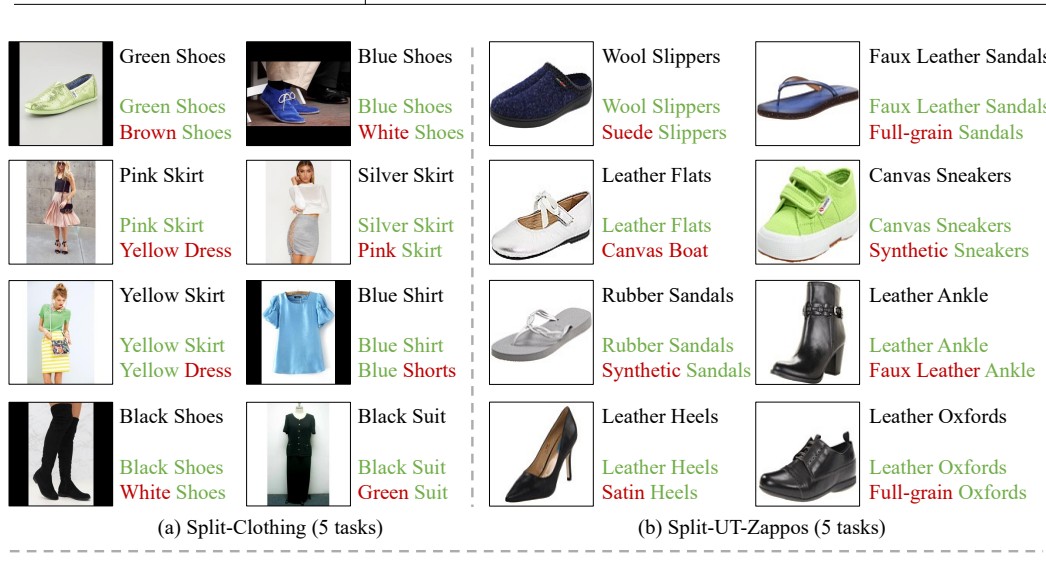

(a) Split-Clothing (5 tasks)    (b) Split-UT-Zappos (5 tasks)

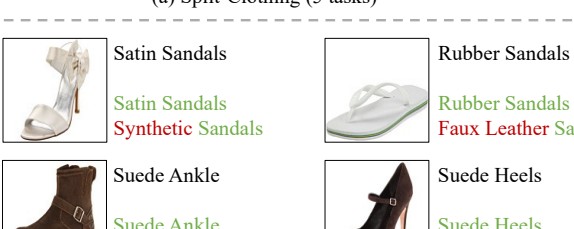

(c) Split-UT-Zappos (10 tasks)

Figure 10: More qualitative results. For each sample, top row is ground-truth label (in black), middle row is CompILer prediction, and bottom row is L2P [43] prediction. The primitives in green and red refer to correct and incorrect predictions.

