# OpenReview forum: "Not Just Object, But State: Compositional Incremental Learning without Forgetting"
_NeurIPS.cc/2024/Conference — NeurIPS 2024 poster_

### Official Review · Reviewer_EVGX · 2024-06-25

**Soundness:** 3
**Presentation:** 3
**Contribution:** 2
**Rating:** 6
**Confidence:** 5

**Summary:**

This paper presents a novel setting of incremental learning, named Compositional Incremental Learning (composition-IL). This setting differs from existing ones as it involves recognizing not only new objects (e.g., a shirt), but also their states (e.g., red) and the resulting compositions (e.g., a red shirt). To address this novel setting, the authors present a prompting-based approach that leverages a pre-trained transformer network to learn the tasks incrementally, thus sharing some similarities with existing methods such as L2P, Dual-Prompt, and CODA-Prompt. Additionally, the authors devise a three-way retrieval mechanism, devoting tailored prompt pools to state, object, and composition-level representation learning. In the experimental section, the authors assess several aspects: the comparison with the state of the art, the benefits of the three-way prompting strategy, the impact of injecting object-level information into the state-level prompt, and the contributions of each involved loss function.

**Strengths:**

1) The paper is clear and fluent. Both the dataset and the model are clearly described.
2) Novelty. To the best of my knowledge, the experimental setting is novel and deserving.
3) While existing prompting strategies often organize the prompt pool in a flat structure, this paper advances the field by leveraging on two concepts: multiple prompts pool related to different concepts/tasks, and possible dependencies between prompts of different concepts.

**Weaknesses:**

**Many hyperparameters/tuning complexity (major)**. From a methodological perspective, the model incorporates several loss terms, each requiring a tailored balancing hyperparameter. Additionally, other factors such as the length of each prompt and the size of the prompt pool must be considered. In truth, the authors report the chosen configuration in their experiments in the second paragraph of Sec. 5.2. As can be observed, there are many hyperparameters, which could complicate the model’s application in real-life scenarios. In this respect, the authors should provide a simplified (yet still reliable) version of their approach, pruning some loss terms and configurations that have a negligible impact on results. Indeed, by examining the results of the ablation studies, it appears that there is room for simplifying the training stage.
Statistical significance (major). Concerning Tables 3 and 4, the results achieved by the final configuration of the Compiler are sometimes very close to the simpler, ablated versions. For example, consider Table 4(a) where 'no injection' is close to 'O->S', and Table 3 where 'C+S' is close to 'C+S+O'. In these cases, it is difficult to conclude whether the benefits are due to effective technical advances or should be ascribed to statistical noise. In this respect, I strongly encourage the authors to perform multiple runs for each experiment and average the final results. This approach would render the comparisons more significant.

**More rehearsal baseline (major)**. I would suggest including in the experimental section the results for DER++ [b] and ER-ACE [c], two straightforward baselines based on rehearsal. As the main contribution of this paper concerns the introduction of a new experimental setting, I believe it is important to provide an extensive evaluation of the existing literature. Moreover, the authors should compare their approach with SLCA [d], a recently introduced baseline for the continual fine-tuning of pre-trained models.
Relation with existing works (major). There is a recent research line on multi-label incremental learning; see [a], one of the most cited papers in the field. In this respect, how does the proposed setting deviate from the multi-label scenario? To justify the introduction of a novel setting, the authors should discuss potential overlaps with existing benchmarks more thoroughly. From a more methodological perspective, instead, I would encourage the authors to provide more references while introducing the learning objectives used in their approach. For example, CODA-Prompt devises orthogonality constraints between prompts, sharing strong similarities with the objectives outlined by Eq. 2 and 3. Moreover, aligning the query and the selected prompts is common in prompting-based approaches (see L2P).

**Lacking ablation studies (minor)**. In Tab. 3, it would be important to see the results of single-pool prompt learning, especially for ‘S’ and ‘O’ (‘C’ has instead been already provided). Indeed, I would guess that a simple strategy based only on object-level prompts would be enough to reach satisfactory performance.

**Clarification requested (minor)**. In Sec. 4.2, the authors discuss an attention-based mechanism to inject object-level cues into the state-level prompt. In this respect, they use three learnable projections, indicated by W_Q, W_K, and W_V. However, as these transformations are continuously updated while tasks progress, did the authors consider that these projections could suffer from catastrophic forgetting? It seems that there are no explicit countermeasures against forgetting regarding these variables, which could hinder the efficacy of the retrieval stage.

**Formatting issue (minor)**. Table 4 appears before Table 3.

[a] Kim, C. D., Jeong, J., & Kim, G. (2020). Imbalanced continual learning with partitioning reservoir sampling. In Computer Vision–ECCV 2020: 16th European Conference, Glasgow, UK, August 23–28, 2020, Proceedings, Part XIII 16 (pp. 411-428). Springer International Publishing.

[b] Buzzega, P., Boschini, M., Porrello, A., Abati, D., & Calderara, S. (2020). Dark experience for general continual learning: a strong, simple baseline. Advances in neural information processing systems, 33, 15920-15930.

[c] Caccia, L., Aljundi, R., Asadi, N., Tuytelaars, T., Pineau, J., & Belilovsky, E. (2021). New insights on reducing abrupt representation change in online continual learning. arXiv preprint arXiv:2104.05025.

[d] Zhang, G., Wang, L., Kang, G., Chen, L., & Wei, Y. (2023). Slca: Slow learner with classifier alignment for continual learning on a pre-trained model. In Proceedings of the IEEE/CVF International Conference on Computer Vision (pp. 19148-19158).

**Questions:**

I have no questions.

**Limitations:**

The limitations are briefly discussed in the supplementary materials.

---

> ### Author Rebuttal · Authors · 2024-08-07
>
> Q1: Many hyperparameters/tuning complexity
>
> A1: Thank you for your constructive suggestions. We consider that assigning weights to different loss terms is a common practice. We highly agree with your suggestion to provide a simplified version of the model. As you suggested, the 'C+S' and 'C+O' in Tab 3 can be viewed as simplified CompILer which are denoted as CompILer-S and CompILer-O for clarity. The whole results are shown in the manuscript appendix Tab 6. Note that the all results are computed by composition pool for fair. The methods significantly lag behind the full CompILer on Split-UT-Zappos because when state features have higher semantics or are visually less apparent, the impact of ambiguous composition boundaries becomes more pronounced. Thus, primitive scores are particularly crucial in such cases. We will continue to explore more simplified models to reduce the number of hyperparameters and present the results in the next version.
>
> Q2: Statistical significance
>
> A2: We conducted the multiple runs with different random seeds as shown in PDF Tab 1. CompILer achieves SOTA results across different random seeds, indicating that the performance improvement is not due to statistical noise. We will extend this setting to all methods and experiments in the next version.
>
> Additionally, we want to elaborate on why 'C+S' is close to 'C+S+O'. The purpose of the multi-prompt pool is to be viewed as a decoupling mechanism. The state disentanglement helps obtain clean state features while implicitly decoupling object and aiding compositional learning. Specifically, the tokens used for composition classification interact with the state prompt during MHA operation, allowing the composition to understand what clean state information looks like, which aids the model's learning of state in the composition branch. Meanwhile, the model realizes information irrelevant to the state prompt is object information. Thus, the classifier becomes better at distinguishing which information is useful for state classification and which is for object classification during classification, achieving a 1+1>2 effect. Thus, using two prompt pools yields impressive experimental results. However, this does not imply that C+S are optimal. Only by explicitly incorporating two prompt pools for state and object can they complement each other better to achieve optimal results.
>
> Moreover, the reason 'no injection' is close to 'O->S' is due to the limitations in prompt selection. Prompt selection follows the key-value mechanism, but one drawback of this paradigm is that it cannot be optimized end-to-end. The key and query are used to select a prompt index from the pool, relying on a second, localized optimization to learn the keys because the model gradient cannot backpropagate through the key/query index selection. Unfortunately, object-injected state prompting also suffers from this separate optimization issue, as it is constrained by the same drawback. This is a common problem in prompt-based continual learning. We plan to explore alternative methods in future work.
>
> Q3: More rehearsal baseline
>
> A3: We have included these baselines in Tab 4 of the PDF, where we demonstrate their performance alongside CompILer. For each method, we allocated a memory buffer size of 800, meaning that each class has 10 samples stored on average. As the results show, CompILer achieves the best performance without any rehearsal.
>
> Q4: Relation with existing works
>
> A4: We recognize that there are significant differences between multi-label IL and composition-IL. In a multi-label scenario, ground truth labels correspond to multiple independent entities within an image, where these labels are independent of each other and lack semantic interactions or fine-grained information. In contrast, the labels in composition-IL consist of a single object combined with its descriptive state. Therefore, composition-IL labels describe a single entity in a more fine-grained fashion rather than multiple entities. To better illustrate the difference, we present a figure in Fig 2 of the PDF. Additionally, we will include references to CODA-Prompt and L2P in the methods section of the next version.
>
> Q5: Lacking ablation studies
>
> A5: We have taken your suggestion, and the results are shown in Tab 5 of the PDF. The first and second rows report the results for state and object classification. An interesting finding is that the accuracy of single primitive classification does not even reach that of using a single pool for composition classification. This is because the experimental setup transitions from the traditional Class Incremental Scenario to a Stochastic Incremental Blurry Task Boundary Scenario (Si-blurry) [1]. Si-blurry faces continuous change of classes between batches leads to intra- and inter-task forgettings, making it difficult for the model to retain previously learned knowledge.
>
> Q6: Clarification requested
>
> A6: The object-injected state prompting module primarily focuses on selecting state prompts to enhance network plasticity and achieve Complementary Learning Systems (CLS). CLS posits that humans continually learn through the synergy of two learning systems: the hippocampus, which specializes in learning pattern-separated representations from specific experiences, and the neocortex, which focuses on acquiring more general and transferable representations from sequences of past experiences. The continuously updated weights Q, K, and V act as a global knowledge transmitter, propagating knowledge from old to new, thereby sharing the knowledge from old to new. Additionally, we would like to emphasize that CompILer addresses catastrophic forgetting by freezing the pre-trained ViT which aligns with findings in [2].
>
> Q7: Formatting issue
>
> A7: Thanks a lot. We will revise their orders.
>
> [1] Online Class Incremental Learning on Stochastic Blurry Task Boundary via Mask and Visual Prompt Tuning
>
> [2] Consistent Prompting for Rehearsal-Free Continual Learning

---

> > ### Comment · Reviewer_EVGX · 2024-08-08
> >
> > I sincerely thank the reviewers for the efforts they spent during the rebuttal period. I still have several considerations and questions for the authors.
> >
> > **Hyperparameters and complexity**. While I acknowledge that tuning several loss coefficients is a standard practice in deep learning, my concerns still persist. Therefore, I urge the authors to reconsider their approach in the final version and, if feasible, reduce the number of terms involved during optimization. Based on the experimental results, there appears to be some flexibility to achieve this reduction. It's important to keep in mind that future users will need to apply this method to new benchmarks, and the abundance of hyperparameters could make this process quite challenging.
> >
> > **Forgetting** In my original review, I expressed some concerns regarding the learnable projections W_Q, W_K, and W_V, which are likely to suffer from catastrophic forgetting as no explicit countermeasures have been taken to protect these parameters. The authors' response, which relies on biological arguments, is somewhat difficult for me to follow. Could the authors provide a more practical discussion on this point? I believe addressing this issue is important.

---

> ### Author Response · Authors · 2024-08-10
>
> We appreciate the reviewer EVGX’s willingness to engage in further discussion. The followings are our point-point answer:
>
> **Q1: About hyperparameters and complexity.**
>
> **A1:** We appreciate your suggestion very much. We thereby have acted on it by developing a simplified model to reduce the number of hyperparameters. Specifically, we focus on the 'C+S+O' configuration, namely Sim-CompILer, which eliminates the directional decoupled loss, RCE loss, and Object-injected State Prompting, accordingly removing $\lambda_1$, $\lambda_2$, and $\alpha$ in the total loss cost. As a result, Sim-CompILer is constrained by vanilla CE loss and a surrogate loss, represented as $L_{total}=L_{CE}+\lambda_3L_{sur}$, where the remaining hyperparameter  is $\lambda_3$ only, which controls the balance between CE and surrogate loss. Apart that, $\beta$, is still needed to adjust a balance between compositions and primitives. The results, shown in the table below, show that, while CompILer performs best across both datasets, Sim-CompILer slightly lags behind but still achieves second-best results compared to the baselines in Table 1 and Table 2. We opted for three pools rather than two due to poor performance with dual prompt pools in Split-UT-Zappos, as noted in our previous rebuttal. We will include the results for this Sim-CompILer alternative in the final version. Hopefully, this simpler alternative would provide more flexibility for the potential users to conduct their related research.
>
> | Name          | Split-Clothing (5 tasks) |   |   |   |   | Split-UT-Zappos (5 tasks) |   |   |   |  |
> |---------------|--------------------------|---|---|---|---|---------------------------|---|---|---|---|
> |               | Avg Acc                  | FTT(↓) | State | Object | HM   | Avg Acc                  | FTT(↓) | State | Object | HM   |
> | Sim-CompILer | 88.22                    | 8.37  | 90.95  | **96.47** | 93.36 | 46.43                    | 19.31  | 56.83  | **79.58** | 66.31 |
> | CompILer      | **88.74**                | **6.98** | **91.61** | 96.34 | **93.92** | **47.06**                | **18.84** | **57.52** | 79.53  | **66.75** |
>
> **Q2: About forgetting**
>
> **A2:** We apologize our explanation confused you due to the complex biological theory. We acknowledge that we have not implemented explicit methods to circumvent catastrophic forgetting, and this choice was intentional for the following reasons:
>
> First of all, following L2P, our model avoids catastrophic forgetting by freezing a ViT backbone across all incremental sessions. Thanks to this simple yet effective strategy, L2P and its improved variants (such as Dual-Prompt, CODA-Prompt, LGCL and our CompILer) achieve much lower forgetting rates than EWC and LwF, as shown in Table 1 of the submitted manuscript. In addition, following the suggestion from Reviewer x9g2, we perform more comparison with NCM and FeCam. While these two methods are not built based on L2P, they still suggest freezing the backbone to overcome the catastrophic forgetting effectively. Inspired by these results, we can note that the core of learning without forgetting relies on a frozen backbone, and a limited number of learnable parameters have less impact on raising the forgetting. This finding is consistent with the fact that L2P did not exploit any explicit countermeasures between the learnable prompts to alleviate the forgetting yet. Akin to the learnable prompts, we advocate learning the parameters in the object-injected state prompting module devoid of extra explicit countermeasures.
>
> On the other hand, we would like to emphasize again that the aim of continual learning is to pursuit an optimal balance between stability and plasticity. Since the stability has been obtained by freezing the backbone, another challenge is how to improve the plasticity when adapting the model on new tasks. Thanks to the learnable parameters in the prompt pool and object-injected state prompting module (even though their number is slighter than that of the backbone), it allows us to improve the adaptability of the model and increase the accuracy further.
>
> Last but not least, the model inclines to forget the compositions rather than objects and states, because the object and state primitives may reappear in new tasks. Hence, we apply object-injected state prompting to the primitive branches, leading to minor forgetting on the compositions.

---

> > ### Comment · Reviewer_EVGX · 2024-08-13
> >
> > The authors have answered my questions. I would like to thank them for their efforts, which address most of my initial concerns. As a result, I will raise my score to weak accept.

---

### Official Review · Reviewer_KbBF · 2024-07-07

**Soundness:** 3
**Presentation:** 3
**Contribution:** 3
**Rating:** 6
**Confidence:** 5

**Summary:**

This work presents a new task called Compositional Incremental Learning (composition-IL).
This new task extends the existing class incremental learning to a more fine-grained scenario for more realistic applications.
This work formulates and designs a new composition-IL benchmark based on Clothing16K and UT-Zappos50K datasets.
Technically, the authors propose several novel strategies based on prompting on pretrained ViT.
Specifically, this work suggests using three prompt pools for states, objects, and their compositions respectively.
To overcome the gap between the pretrained task and the composition task, the object prompts are further injected to guide the selection of state prompts.
The authors also suggest a learnable generalized-mean prompt fusion scheme for prompt reducing.
Extensive experiments are provided to evaluate the models.

**Strengths:**

1. This paper has done a great job in presentation, with clear wiring and figuring, enabling the fast understanding of reviewers familiar with continual learning.
2. The contribution of proposing novel and more realistic tasks is always welcomed. And this work has done some essential initial jobs such as creating a benchmark for composition-IL.
3. The proposed method looks convincing to me.

**Weaknesses:**

1. The main concern for me is the proposed composition-IL itself. From the experimental results (Tables 1 and 2), it seems that the previous prompt-based works can also handle this task with comparable performance. So my double is that if this work overclaims the difficulty of composition-IL? As someone working in the continual learning field, I think new settings and tasks are always welcomed only if the new task is challenging and practical enough.
2. The effectiveness of the proposed components. As shown in Table 3, I think the performance gap between C+S+O and C+S/C+O is small, so the effectiveness of multi-pool prompt learning is not that significant. Similar results are also in Table 4. In Table 4(a), there is only a 0.32% improvement in HM when introducing the object injection. In Table 4(b), there is also only a 0.3% improvement when using the GeM instead of the naive mean pooling.

**Questions:**

Please see the 'Weakness' section for detailed questions.
In conclusion, I expect the authors to provide their thoughts about the necessity of the proposed composition-IL setting, as well as some explanations of the small performance gap when equipped with the proposed modules.

**Limitations:**

The authors have adequately addressed the limitations and potential negative societal impact of their work.

---

> ### Author Rebuttal · Authors · 2024-08-06
>
> Q1: About the composition-IL
>
> A1: Thank you for your great comment. Firstly, we would like to emphasize that previous prompt-based works have achieved satisfactory results by employing strategies that contradict the paradigm of continual learning. The compared prompt-based methods all involve task-specific parameters except L2P, which means that their network parameters increase gradually with the introduction of more incremental tasks. This approach effectively enhances the performance but goes against the fundamental principles of continual learning, which aims for a fixed model to incrementally learn new tasks while overcoming forgetting of old tasks. The training paradigm of increasing parameters clearly contradicts this requirement. These competitive results achieved by network expansion should not decrease the awareness of significance of composition-IL.
>
> Instead of network expansion, L2P shows a large gap compared to the Upper Bound, as evidenced in Table 1 of the paper, demonstrating the significant necessity for further research. L2P is the only method in previous work that does not employ the aforementioned tricks. It shows a considerable performance collapse across three settings compared to One-batch learning, failing to even reach 50% of the Upper Bound. This significant degradation underscores the challenges faced by composition-IL due to ambiguous composition boundaries, highlighting the continued necessity for research. Hence, exploring methods that strictly adhere to the continual learning paradigm and achieve satisfactory results remains highly demanded and challenging.
>
> Furthermore, we underscore the research significance of composition-IL from a theoretical perspective. The hallmark of the human cognitive system is compositionality, allowing people to decompose and recombine knowledge to better understand and facilitate learning of new knowledge. Therefore, much scholarly attention has focused on how to equipe the model with compositionality. Unfortunately, there is little focus on studying compositionality in incremental learning scenarios. We recognize that in many real-world scenarios, decisions often rely on state-object pairs. Therefore, we propose Composition-IL to address the gap in modeling states within incremental learning. For instance, in autonomous driving systems, decisions often hinge not just on detecting pedestrians but also on understanding their state to determine the vehicle's next actions. Hence, this task is highly practical and worthy of pursuit.
>
> Q2: The effectiveness of the proposed components.
>
> A2: About C+S+O: The purpose of the multi-prompt pool can be seen as a decoupling mechanism. Compared to a single prompt pool, the three-prompt pool exhibits a significant improvement, increasing performance by 8.73%. Introducing an additional pool (decoupling a single primitive), can lead to substantial performance gains. This is because arbitrary primitive disentanglement helps to obtain clean, independent primitive features while implicitly decoupling another one and aiding compositional learning. For example, when using two prompt pools to separately learn composition and state, the state prompt and composition prompt are concatenated together with feature sequence and processed through MHA. During this process, the tokens used for composition classification interact with the state prompt, allowing the composition to understand what clean state information looks like, which aids the model's learning of state in the composition branch. Simultaneously, the model recognizes information irrelevant to the state prompt as object information. As a result, during classification, the classifier performs better at distinguishing which information is useful for state classification and which is for object classification, achieving a 1+1>2 effect. Therefore, using two prompt pools yields impressive experimental results. However, this does not imply that two pools are necessarily optimal. Only by explicitly incorporating two prompt pools for state and object can they complement each other better to achieve optimal results, as shown in manuscript Table 3.
>
> About Object-injected State Prompting: The expected improvement from Object-injected State Prompting is not as apparent due to limitations in the prompt selection mechanism. Like L2P, prompt selection follows the key-value mechanism. A drawback of this paradigm is that it cannot be optimized in an end-to-end fashion because it uses keys and queries to select a prompt index from the pool. This reliance on a second, localized optimization to learn the keys means that the model gradient cannot backpropagate through the key/query index selection, leading to only minor improvements in the key. Unfortunately, injected state prompting also suffers from this limitation, as it is subject to separate optimization without model gradient improvements. This is a common issue in prompt-based continual learning and is worth exploring further. We plan to investigate alternative methods in future work.
>
> About Generalized Mean Pooling: GeM can be seen as a balance between Max Pooling and Mean Pooling. We have analyzed it from the derivatives view in our appendix. We leverage GeM to filter out irrelevant information from the selected prompts. As a result, the fewer prompts selected, the higher relevance between these prompts and the images. Therefore, the effectiveness of GeM is linearly related to the amount of information to be integrated. The more information that needs to be fused, the more pronounced the effect of GeM Pooling.

---

> > ### Comment · Reviewer_KbBF · 2024-08-12
> >
> > I thank the authors for their rebuttal.
> > The rebuttal solved part of my concerns, so I will keep my score unchanged.

---

### Official Review · Reviewer_B271 · 2024-07-07

**Soundness:** 3
**Presentation:** 3
**Contribution:** 3
**Rating:** 6
**Confidence:** 4

**Summary:**

This paper introduces a novel task termed Compositional Incremental Learning (composition-IL), which aims to enable models to recognize a variety of state-object compositions incrementally. The authors propose a new model called CompILer, which employs multi-pool prompt learning, object-injected state prompting, and generalized-mean prompt fusion to overcome the challenges in this task. The study utilizes two newly tailored datasets, Split-Clothing and Split-UT-Zappos, and demonstrates state-of-the-art performance through extensive experiments.

**Strengths:**

1. The introduction of composition-IL addresses a critical gap in incremental learning by focusing on the recognition of state-object compositions, which traditional class-IL and blur-IL approaches overlook
2.  The proposed CompILer model is well-conceived, leveraging multi-pool prompt learning, object-injected state prompting, and generalized-mean prompt fusion to enhance the learning of compositions.

**Weaknesses:**

1. The paper shows that CompILer only slightly outperforms LGCL and CODA-Prompt on the Split-Clothing and Split-UT-Zappos datasets. However, the authors do not detailedly analyze why LGCL and CODA-Prompt perform well in these tasks or why CompILer's improvements are limited. A detailed comparative analysis is needed.

2. Lack of analysis on model efficiency, especially on the size of learnt prompts comparing to current prompt-based methods

3. The paper lacks a systematic analysis of the hyperparameters $\lambda_1$ and $\lambda_2$ in Equation 9.

**Questions:**

Model Efficiency: How does the size of the learnt prompts in CompILer compare to current prompt-based methods? Please provide a detailed analysis of model efficiency.

Hyperparameter Analysis: Can you provide a systematic analysis of the hyperparameters $\lambda_1$ and $\lambda_2$ in Equation 9? How do variations in these values impact the model's performance?

Performance Comparison: Why do LGCL and CODA-Prompt perform well on the Split-Clothing and Split-UT-Zappos tasks, and what factors limit the performance improvements of CompILer over these methods? A detailed comparative analysis would be beneficial.

**Limitations:**

The authors have acknowledged the limitations of their work, particularly concerning the performance challenges on the Split-UT-Zappos dataset due to long-tail distributions and the difficulty in distinguishing state classes.

---

> ### Author Rebuttal · Authors · 2024-08-06
>
> Q1: About performance comparison.
>
> A1: Thanks for your question. We are glad to answer it. First, we would like to emphasize that CompILer has achieved SOTA performance across all settings. Notably, on the Split-Clothing dataset, CompILer beats LGCL and CODA-Prompt with an improved average accuracy of 1.55% and 1.63%, respectively. The reason why the performance improvement might not meet the reviewer's expectations is that both LGCL and CODA-Prompt rely on parameter expansion training paradigms. As what we have mentioned in line 263 of our paper, the parameters of these two methods increase significantly when more new tasks are incoming. Although introducing task-specific parameters undoubtedly boosts the performance, this approach contradicts the core principle of continual learning, where the network parameters should not increase unlimitedly as the number of tasks grows. Additionally, as noted in line 255 of our paper, LGCL introduces external semantic priors to achieve language guidance. Although this trick can further enhance the performance, it also limits the model's applicability. For instance, LGCL fails to operate on the 5-task Split-UT-Zappos since the total length of class names exceeds the limitation. In contrast, CompILer does not employ any parameter-increasing techniques or additional semantic knowledge. Compared to our baseline L2P, which does not use parameter expansion, CompILer shows a significant improvement, with 8.73%, 4.91%, and 3.13% across the three settings. We will shed more light on the comparison in the next version. We will emphasize this point again in the camera-ready version and expand on this section.
>
> Q2: About model efficiency.
>
> A2: We have followed your suggestion and further compared the size of prompts used in CompILer and other prompt-based methods. As shown in Tab. 3 of the attached PDF, the size of the prompt pool in CompILer is smaller than that in CODA-Prompt but is larger than that in Dual-Prompt and LGCL. As anticipated, CompILer demonstrates higher model efficiency for a longer sequence of tasks, because we will not dynamically increase the model parameters when we have pre-define the size of prompt pool. Consequently, in such scenarios, we can conjecture that CompILer’s memory footprint and inference speed are more efficient compared to Dual-Prompt and CODA-Prompt.
>
> Q3: Lack of hyperparameter analysis
>
> A3: We have added more experiments regarding the parameters $\lambda_1$ and $\lambda_2$, as shown in Fig. 1 of the attached PDF. The model's average accuracy shows an initial increase followed by a decrease within the interval, reaching a peak at $\lambda_1=1.0$ and $\lambda_2=3e-5$. Therefore, we consider the model to be at a local optimum in this setting and choose these parameters as the final setting.

---

> > ### Comment · Reviewer_B271 · 2024-08-13
> >
> > Dear Authors,
> >
> > Thank you for your clear responses and additional analyses. I appreciate your work and believe it adds valuable contributions to the field. I raise my rating to 6.

---

### Official Review · Reviewer_x9g2 · 2024-07-11

**Soundness:** 3
**Presentation:** 3
**Contribution:** 3
**Rating:** 6
**Confidence:** 4

**Summary:**

The paper propose compositional Incremental Learning to enable models to recognize state-object compositions incrementally. The paper provides two tailored datasets for composition-IL by modifying two existing datasets in the fashion domain. The paper propose a new prompt-based model comprising of multi-pool prompt learning, object-injected state prompting and generalized-mean prompt fusion. The method achieves competitive performance on the two proposed datasets.

**Strengths:**

1. The paper discusses a new direction which aims to exploit state primitives of objects for incremental classification. The authors propose a new incremental setting where objects or states can reappear in new classes in new tasks.
2. The paper provides two curated datasets in the fashion domain to study composition-IL.
3. The paper propose a new prompt-based model for composition-IL. It is nice to report HM accuracy for better evaluation.

**Weaknesses:**

I have major concerns with the experimental part of the paper.
1. Hyperparameters: It is very concerning that six hyperparameters are tuned on the testing set. The hyperparameter values even differ for different task splits on the same dataset. It is weird that the authors use 25 epochs for one dataset, while for the other dataset, 10 epochs for 5-task setting while 3 epochs for the 10 task setting. Even different learning rates are used for different settings on same datasets. How are these decided? It looks like everything is optimized for the test sets in all settings. This is not a fair way of doing experiments. It is acceptable if the authors fine-tune the model for one dataset and use the same parameters for all settings across different datasets (I think this is commonly done in continual learning domain). How can the proposed method be useful/practical if it needs to fine-tune so many hyper parameters on every setting using the test set to get good results?
2. Lack of experiments with random seeds: The experiments are conducted using a single random seed. The proposed method has improvements of 1% or even less in some settings. It is standard practice in CL to report results with multiple random seeds for fair evaluation and to establish the robustness of the model.
3. Competitive recent baselines like HiDe-prompt [1] are not included in the comparison.
4. Simple methods like NCM classifier [2] and Mahalanobis-distance based classifier [3] outperforms prompt methods like L2P on several datasets with first-task adaptation and no training in new tasks (using frozen model after the first task and doing continual evaluation). It would be interesting to see how these strong baseline methods work in the proposed settings.

[1] Wang, Liyuan, et al. "Hierarchical decomposition of prompt-based continual learning: Rethinking obscured sub-optimality." Advances in Neural Information Processing Systems 36 (2023).

[2] Paul Janson,  et al, A simple baseline that questions the use of pretrained models in continual learning. arXiv preprint arXiv:2210.04428, 2022.

[3] Dipam Goswami, et al, Fecam: Exploiting the heterogeneity of class distributions in exemplar-free continual learning. In Thirty-seventh Conference on Neural Information Processing Systems, 2023.

**Questions:**

1. I am curious what is the difference between object-state pairs and a group of concepts describing a class? Like why limit the concepts to just object and state, there can be more concepts attached to a class. So, a class can also be described as a group of concepts and classes can then be learned incrementally with overlapping concepts from old classes.

**Limitations:**

Limitations of this work is not explicitly addressed in paper.

---

> ### Author Rebuttal · Authors · 2024-08-07
>
> Q1: About hyperparameter tuning
>
> A1: Thanks for your question. Firstly, we claim that finding the optimal balance between hyperparameters is important yet challenging for continual learning. Since the feature distributions within each incremental session may vary considerably, the optimal hyperparameters for each session are often different. Fortunately, our work focuses on using a set of hyperparameters across all sessions, rather than tuning different hyperparameters for each individual session.
>
> Moreover, we state that due to the complexity of the dataset and the differences in feature distributions, different hyperparameters are often assigned to different datasets. We find this phenomenon is quite common in prompt-based continual learning. For example, Dual-Prompt trains CIFAR-100 with a learning rate of 0.03 for 5 training epochs, whereas it trains ImageNet-R with a learning rate of 0.005 for 50 epochs. Likewise, LGCL's class-level and task-level objective weights on CIFAR-100 are 0.137 and 0.323, respectively, accurate to even three decimal places.
>
> Furthermore, for the same dataset, different settings are often assigned with different hyperparameters to ensure the optimal performance. It is a common strategy in compositional learning. For example, in the milestone work Co-CGE [1] in Compositional Zero Shot Learning (CZSL), the method trains for 300 epochs in a closed-world scenario on the CGQA dataset but only for 200 epochs in an open-world scenario. We emphasize that in CZSL, the training data for both open-world and closed-world scenarios is exactly the same, which is essentially no different from our case of hyperparameter setting. In compositional filed, the reason that different hyperparameters can be assigned for the same dataset under different settings is due to the complex interplay of the feature space resulting from compositionality; without appropriate hyperparameters, the model’s potential may be severely limited. Our empirical choice of different hyperparameters is based on: for Split-Clothing, with its simpler primitive description, a larger number of epochs should be chosen to ensure learning a more accurate representation, whereas due to the visual invisibility of the state in Split-UT, excessively large epochs may lead to overfitting on some irrelevant information. For example, the model might mistakenly associate “Suede” with “black” because many “Suede shoes” in the dataset appear in black.
>
> Finally, the core contributions and innovations of our work lie in the novel task setting, two corresponding datasets, and the method for solving ambiguous composition boundaries. We acknowledge that setting the same parameters across all datasets is an optimal setup as it demonstrates a model's generalizability, but this is not the central problem our work addresses. However, we will also consider the reviewers' feedback and explore a more robust model in future work, which will be discussed in the next version.
>
> Q2: Lack of experiments with random seeds
>
> A2: We greatly appreciate the reviewer's reminder. We conducted experiments with 4 different random seeds as shown in Tab 1 of the PDF. Although there are slight fluctuations in the results, it consistently achieves SOTA performance across all metrics. Although single seed experiments are acceptable in recent continual learning works [2][3], we will use multiple random seeds for a more thorough comparison.
>
> Additionally, we clarify that CompILer does not use any extra semantic knowledge, does not store any information of old session, does not allocate any task-specific parameters, and does not dynamically increase network parameters. We believe that these techniques (which are commonly used by prompt-based methods except L2P) undoubtedly contribute significantly to performance improvements, but they are contrary to the principles of continual learning, as noted in line 263 of our manuscript.
>
> Q3: Lack of methods
>
> A3: We have conducted comparisons with the methods you suggested, and the results are shown in Tab 2 of the PDF. Due to limited rebuttal period, we mainly compare Compiler with the methods on Split-UT-Zappos. HiDe-Prompt surpasses CompILer due to the task-specific prompt settings and the storage of statistics of old classes which seem to contradict the principles of continual learning to some extent.
> For NCM and Fecam, unfortunately, although this first-task adaptation approaches achieve a lower forgetting rate, their plasticity on incremental tasks is quite poor because they only work well when the first task contains the most classes. We will include these methods as baselines in the next version and provide proper citations.
>
> Q4: The relationship with a group of concepts.
>
> A4: Some datasets associate classes with a group of concepts. For example, in the CUB dataset, a number of attribute concepts are provided for classes, such as the color of the leg/back, the pattern of the head/breast, etc. However, these attribute information is locally semantic, and cannot encapsulate the global state features of a class. As a result, it limits the attribute information forming a pair with the object. In contrast, the composition-IL task combines objects with global states, so that objects and states are considered on an equal status and exhibit compositionality. Notably, the original datasets Clothing16K and UT-Zappos50K only provide a single state description for each class which prevents Split-Clothing and Split-UT from forming an object with a group of global concepts. Nevertheless, we also acknowledge your advice and it is potential to explore a new benchmark with multiple global states further.
>
> [1] Learning Graph Embeddings for Open World Compositional Zero-shot Learning (TPAMI 2022)
>
> [2] When Prompt-based Incremental Learning Does Not Meet Strong Pretraining (ICCV 2023)
>
> [3] Fecam: Exploiting the heterogeneity of class distributions in exemplar-free continual learning (NeurIPS 2023)

---

> > ### Comment · Reviewer_x9g2 · 2024-08-10
> >
> > I appreciate the detailed rebuttal for the authors which answers most of my concerns.
> > 1. Hyperparameters: While the authors argue that several papers use different training hyper-parameters for different settings even using the same dataset, I still believe that this is not a robust solution, it is just optimizing the results on the test set which does not make sense. Just saying that some existing papers do the same overfitting on the test set is not enough. I would urge the authors to reduce the number of hyper-parameters (following the discussion with Reviewer EVGX) to make the method usable and more realistic. Improving the accuracy by small margins is not everything, what's more important is how robust and practical your solution is.
> > 2. Comparison with baselines: It is appreciable that the authors reproduced the baseline methods. HiDE-prompt was expected to do better, it's good to have a discussion in the paper about the trade-offs. NCM is a baseline which can be considered as a lower bound since it does not update the model. FeCam is naturally expected to do better than NCM since it uses Mahalanobis-distance and has improved over NCM using ViTs [3]. So, it's quite surprising to see that it gets worse. I would ask the authors to look into the implementation or find appropriate justifications for this since FeCam results in Table 2 looks like an error or faulty implementation. It is important to discuss these baselines since they form the lower bounds and shows the scope of improvement using the prompt methods.
> >
> > I still maintain the concerns for the hyper parameters and urge the authors to look into this. I raised my rating to 5.

---

> ### Author Response · Authors · 2024-08-13
>
> Thank you very much for further discussion and comments. We appreciate very much for your raising the rating. We are happy to answer your questions below:
>
> **Q1: About hyperparameters.**
>
> **A1:** We have acted on your suggestion to reduce the number of hyperparameters by streamlining the model. Specifically, we implement a simplified alternative model (namely Sim-CompILer) , which removes the directional decouple loss, RCE loss, and Object-injected State Prompting from the original CompILer, thereby eliminating hyperparameters $\lambda_1$, $\lambda_2$, and $\alpha$, Note that, $\lambda_3$ still remains in order to balance CE loss and surrogate loss, and $\beta$ is still needed to adjust the trade-off between compositions and primitives. The results, shown in the table below, demonstrate that, even though CompILer achieves the best performance across both datasets, Sim-CompILer slightly underperforms but still ranks second. We agree with your suggestion and believe this simplified version is more practical and easy-to-follow. We will include the results for Sim-CompILer in the final version. Hopefully, this simpler alternative would provide more flexibility in potential practical applications.
>
> | Name          | Split-Clothing (5 tasks) |   |   |   |   | Split-UT-Zappos (5 tasks) |   |   |   |  |
> |---------------|--------------------------|---|---|---|---|---------------------------|---|---|---|---|
> |               | Avg Acc                  | FTT(↓) | State | Object | HM   | Avg Acc                  | FTT(↓) | State | Object | HM   |
> | Sim-CompILer | 88.22                    | 8.37  | 90.95  | **96.47** | 93.36 | 46.43                    | 19.31  | 56.83  | **79.58** | 66.31 |
> | CompILer      | **88.74**                | **6.98** | **91.61** | 96.34 | **93.92** | **47.06**                | **18.84** | **57.52** | 79.53  | **66.75** |
>
> **Q2: Comparison with baselines.**
>
> **A2:** Thank you for pointing out this issue. After checking our experiments, we confirmed that there was an error in the implementation of NCM and FeCAM. Concretely speaking, FeCAM provides two official implementations, one under the Avalanche Library [1] and the other under PyCIL [2]. Since the Avalanche Library includes both NCM and FeCAM, we conducted our experiments within this framework to ensure a fair comparison. Unfortunately, during the process, we overlooked the need to update the class mask, determining which classes have been seen during training the model incrementally. This negligence leads to the unreasonable results you have noticed. Also, we made the same mistake when implementing the NCM method, which also obtained unsatisfactory results. To correct the error, we have fixed the mask and re-conducted the experiments from scratch. The new results, as shown in the following table, verify the fact that FeCAM should achieve higher accuracy than NCM.
>
> CompILer outperforms NCM and FeCAM by enhancing adaptation through the tuning of learnable prompts across all sessions. In contrast, NCM and FeCAM are limited by their inability to update the model during incremental learning. Additionally, it is noteworthy that CompILer significantly surpasses NCM and FeCAM in Object accuracy. This is because CompILer freezes the backbone across all sessions and leverages the pre-trained parameters, while the other two methods update the backbone in the first session, which weakens their effectiveness in object recognition. We will include these methods as baselines in the next version and provide proper citations.
>
> | Method   | Prototype | Avg Acc | State  | Object | HM     |
> |----------|-----------|---------|--------|--------|--------|
> | NCM      | ✓         | 31.09   | 41.91  | 39.71  | 40.78  |
> | FeCAM    | ✓         | 33.71   | **46.32** | 40.44  | 43.18  |
> | CompILer | ✗         | **34.66** | 45.82  | **77.06** | **57.47** |
>
> [1] Avalanche: an End-to-End Library for Continual Learning (CVPRW 2021)
>
> [2] Pycil: a python toolbox for class-incremental learning (SCIENCE CHINA Information Sciences 2023)

---

> > ### Comment · Reviewer_x9g2 · 2024-08-13
> >
> > I appreciate the authors efforts in improving the paper and clarifying most of my concerns.
> >
> > It is good that the authors propose a simpler version of the method which still works with lesser hyper-parameters and should be included in the paper. I expect the authors to have the discussion in the paper on the need to reduce the hyper-parameters and use simpler models.
> >
> > It is interesting to see that the baselines of NCM and FeCam performs very competitively with the prompt-based methods and is better in predicting the state while worse in object prediction. It is good to have a detailed discussion on comparison with these frozen-model baselines which highlights how the learnable prompt-parameters (also using frozen model) are affecting the performance. It is also good to add them in the model efficiency analysis (like in Table 3 rebuttal) since they do not learn any new parameters.
> >
> > Overall, I am now quite positive about the contribution of this work which opens up a very interesting direction in Composition-IL with very intuitive and novel setting, method, datasets and very good analysis. The extensive experiments (including the baselines and also more rehearsal-based method comparisons added during rebuttal) makes it a good benchmark for future works.
> >
> > One minor remark for the final version would be to add proper informative captions for all figures in the paper which now is quite empty, it makes the paper more readable and easy to understand. I raise my rating to 6.

---

### Author Rebuttal · Authors · 2024-08-07

We sincerely thank the reviewers x9g2 (R1), B271 (R2), KbBF (R3), and EVGX (R4) for their constructive comments and acknowledgements: “the proposed new task composition-IL is novel and welcome”(R1, R2, R3, R4); “the proposed benchmarks are well-constructed”(R1, R3); “CompILer is well-conceived”(R2, R3); “the paper is good presentation”(R3, R4); “the metric is meaningful”(R1).

---

### Decision · Program_Chairs · 2024-09-25

**Decision:**

Accept (poster)

**Comment:**

After the rebuttal and the authors-reviewers discussion, all reviewers recommended acceptance of the work. After reading the paper, the reviews, and the discussion, I see no reason to overturn their assessment, as the proposed approach and problem formulation can inspire future research in the intersection between compositionality and continual learning. The authors are encouraged to include and discuss the promised baselines and results (e.g., multiple seed runs, Sim-CompILer, additional baselines from the pdf), and follow x9g2's suggestions on captions and hyperparameters discussion.